# Suppressing mosquito populations with precision guided sterile males

Ming Li [1,9], Ting Yang [1,9], Michelle Bui[1], Stephanie Gamez [1], Tyler Wise[1], Nikolay P. Kandul[1], Junru Liu [1], Lenissa Alcantara[1], Haena Lee[1], Jyotheeswara R. Edula[1,2,8], Robyn Raban[1], Yinpeng Zhan[3], Yijin Wang[3], Nick DeBeaubien[3], Jieyan Chen[3], Héctor M. Sánchez C.[4], Jared B. Bennett[4,5], Igor Antoshechkin[6], Craig Montell [3], John M. Marshall [4,7] & Omar S. Akbari [1,2✉]

The mosquito *Aedes aegypti* is the principal vector for arboviruses including dengue/yellow fever, chikungunya, and Zika virus, infecting hundreds of millions of people annually. Unfortunately, traditional control methodologies are insufficient, so innovative control methods are needed. To complement existing measures, here we develop a molecular genetic control system termed precision-guided sterile insect technique (pgSIT) in *Aedes aegypti*. PgSIT uses a simple CRISPR-based approach to generate flightless females and sterile males that are deployable at any life stage. Supported by mathematical models, we empirically demonstrate that released pgSIT males can compete, suppress, and even eliminate mosquito populations. This platform technology could be used in the field, and adapted to many vectors, for controlling wild populations to curtail disease in a safe, confinable, and reversible manner.

[1] Division of Biological Sciences, Section of Cell and Developmental Biology, University of California, San Diego, La Jolla, CA, USA. [2] Tata Institute for Genetics and Society, La Jolla, CA, USA. [3] Department of Molecular, Cellular, and Developmental Biology and the Neuroscience Research Institute, University of California, Santa Barbara, CA, USA. [4] Divisions of Epidemiology & Biostatistics, School of Public Health, University of California, Berkeley, CA, USA. [5] Biophysics Graduate Group, University of California, Berkeley, CA, USA. [6] Division of Biology and Biological Engineering (BBE), California Institute of Technology, Pasadena, CA, USA. [7] Innovative Genomics Institute, University of California, Berkeley, CA, USA. [8] Present address: Tata Institute for Genetics and Society (TIGS), TIGS Center at inStem, GKVK Campus, Bangalore, Karnataka, India. [9] These authors contributed equally: Ming Li, Ting Yang. ✉email: oakbari@ucsd.edu

Mosquitoes are the world's deadliest animals, killing more humans on earth than any other animal[1]. They transmit the majority of vector-borne diseases, such as the notorious arboviruses dengue, Zika virus, yellow fever, and chikungunya transmitted by *Aedes* mosquitoes. The predominating strategy to control these devastating diseases is the use of insecticides, though mosquitoes are evolving and spreading insecticide resistance[2], hampering control efforts. Therefore, there is an urgent demand for innovative mosquito-control technologies that are effective, sustainable, and safe.

Alongside traditional control measures, several genetic-based techniques are being used to combat mosquitoes. These include multiple male (♂) release programs aimed at population suppression, such as the classical radiation-based sterile insect technique (SIT), relying on releasing irradiated sterile ♂'s (plural)[3]. Alternative approaches include the *Wolbachia*-based incompatible insect technique (IIT), relying on the release of *Wolbachia*-infected ♂'s[4–6], or the antibiotic-based release of insects carrying a dominant lethal gene (RIDL)[7]. Moreover, emerging CRISPR-based homing gene drives that spread target genes through a population faster than through traditional Mendelian inheritance are presently under development with the aim of safe implementation in the future[8–10].

As an alternative, a CRISPR-based technology termed precision guided SIT (pgSIT), was recently innovated[11]. pgSIT uses a binary approach to simultaneously disrupt genes essential for female (♀) viability and ♂ fertility, resulting in the exclusive survival of sterile ♂'s that can be deployed at any life stage to suppress and eliminate populations. It requires two breeding strains, one expressing Cas9 and the other expressing guide RNAs (gRNAs). Mating between these strains results in RNA-guided mosaic target gene mutations throughout development, ensuring complete penetrance of desired phenotypes. Compared to alternatives, pgSIT does not require the use of radiation, *Wolbachia*, or antibiotics, and will not persist in the environment longterm. Unfortunately, this technology is presently only accessible in flies, and for population control techniques, its equivalent needs to be developed for mosquitoes.

To address this need, here we systematically engineer pgSIT in *Ae. aegypti* using a system that simultaneously disrupts genes essential for ♂ fertility and ♀ flight, which is essential for mating, blood feeding, reproduction, and predator avoidance—meaning survival in general[12]. Using our technology, we demonstrate that generated sterile ♂ progeny can compete, suppress and even eliminate mosquito populations in multigenerational population cages when released as either eggs or adults. Mathematical models suggest that releases of *Ae. aegypti* pgSIT eggs could effectively eliminate a local *Ae. aegypti* population using achievable release schemes. Taken together, this study suggests that pgSIT may be an efficient technology for mosquito population control and introduces the first pgSIT system suited for real-world release.

## Results

**Validation of pgSIT target genes**. To engineer pgSIT in *Ae. aegypti*, we first validated target genes by generating transgenic gRNA-expressing lines targeting two conserved genes: β-Tubulin 85D (*βTub*, AAEL019894), specifically expressed in mosquito testes[13–15] and essential for spermatogenesis and ♂ fertility[16], and myosin heavy chain (*myo-fem*, AAEL005656), expressed nearly exclusively in ♀ pupae[13,14] and essential for ♀ flight[17] (Supplementary Fig. 1 and Supplementary Table 1). To ensure efficient disruption, each gRNA line encoded four U6–promoter-driven[18] gRNAs targeting unique sites in the coding sequence of either *βTub* (U6-gRNA^*βTub*—marked with *3xP3-GFP*) or *myo-fem* (U6-gRNA^*myo-fem*—marked with *3xP3-tdTomato*) (Supplementary

Figs. 2–4). Multiple independent transgenic lines were generated given that position effects are expected to result in a variable expression[19,20]. To assess the activity of each line independently, we conducted bidirectional crosses with Cas9 controlled by a homozygous *nuclear pore complex protein* (*Cas9*—marked with *Opie2-CFP*)[21] (Supplementary Tables 2 and 3 and Supplementary Figs. 2–4). The resulting transheterozygous F_1 progeny (gRNA/+, Cas9/+) were assessed and crossed to wildtype (WT) for further evaluation. For the *βTub* crosses, the fecundity of the F_1 transheterozygous ♂'s ranged from 0 to 94.9%, with two lines out of ten demonstrating sterility resulting from immotile sperm[16], while F_1 transheterozygous ♀'s maintained normal fertility (Fig. 1, Supplementary Fig. 2, Supplementary Table 3, and Supplementary Video 1). For *myo-fem* crosses, all F_1 transheterozygous ♀'s generated from three out of five lines were flightless, while F_1 transheterozygous ♂'s maintained normal flight (Fig. 1, Supplementary Table 3, Supplementary Fig. 3, and Supplementary Videos 2 and 3). As expected, ♀ flightlessness significantly reduced mating ability, and blood consumption, as many flightless ♀'s get trapped on the water surface following eclosion, resulting in reduced fertility, fecundity, and survival. Sanger sequencing of genomic DNA revealed expected mutations at the *βTub*- and *myo-fem*-targeted loci.

**Development of pgSIT and fitness assessments**. To generate a pgSIT strain capable of targeting both *βTub* and *myo-fem* simultaneously, we combined two gRNA lines that exclusively produced sterile ♂'s (gRNA^*βTub*#7) or flightless ♀'s (gRNA^*myo-fem*#1) (Fig. 1, Supplementary Figs. 2 and 3, and Supplementary Table 3) by repeated backcrossing, generating a double-homozygous stock (termed gRNA^*βTub + myo-fem*) (Supplementary Fig. 4). To assess its activity, we bidirectionally crossed gRNA^*βTub + myo-fem* to Cas9. Importantly, these crosses yielded all flightless ♀'s (termed pgSIT^♀) and sterile ♂'s (termed pgSIT^♂) with normal flight and mating capacity (Fig. 2, Supplementary Fig. 5, Supplementary Tables 4–8, and Supplementary Videos 4–6). We next determined transgene integration sites, single copy number per transgene, and confirmed target gene disruptions by both amplicon sequencing (Supplementary Fig. 6) and Nanopore genome sequencing using transheterozygous pgSIT^♂'s (Supplementary Figs. 7–9 and Supplementary Tables 9 and 10). We also performed transcriptome sequencing of pupae comparing pgSIT^♂'s and pgSIT^♀'s to WT to quantify target gene reduction, expression from transgenes, and to assess global expression patterns (Supplementary Figs. 8–10 and Supplementary Tables 11–15). As expected, we observed significant target gene disruption in pgSIT individuals, robust expression from our transgenes, and non-target gene misexpression, which would be expected given the significant phenotypes observed (i.e., flightless ♀'s and spermless ♂'s).

To explore potential fitness effects, we assayed several fitness parameters including ♀ fecundity, fertility, flight activity, ♂ mating capacity, ♂ sound attraction, larva–pupa development time, pupa-adult development time, and longevity (Fig. 2, Supplementary Fig. 5, Supplementary Tables 5–8, and Supplementary Videos 5 and 6). The pgSIT^♀'s were flightless with significantly reduced fecundity, fertility, and survival, indicating they would be very unlikely to survive in the wild, let alone transmit pathogens. For pgSIT^♂'s, other than slightly delayed larva–pupa development time, we did not detect significant differences in fitness parameters. Previous studies demonstrated that *Ae. aegypti* ♀'s typically mate only once in their lifetime, a behavior known as monandry[22]. To explore whether prior matings with pgSIT^♂'s could suppress ♀ fertility, we initiated experiments in which WT ♀'s were first mated with pgSIT^♂'s for a

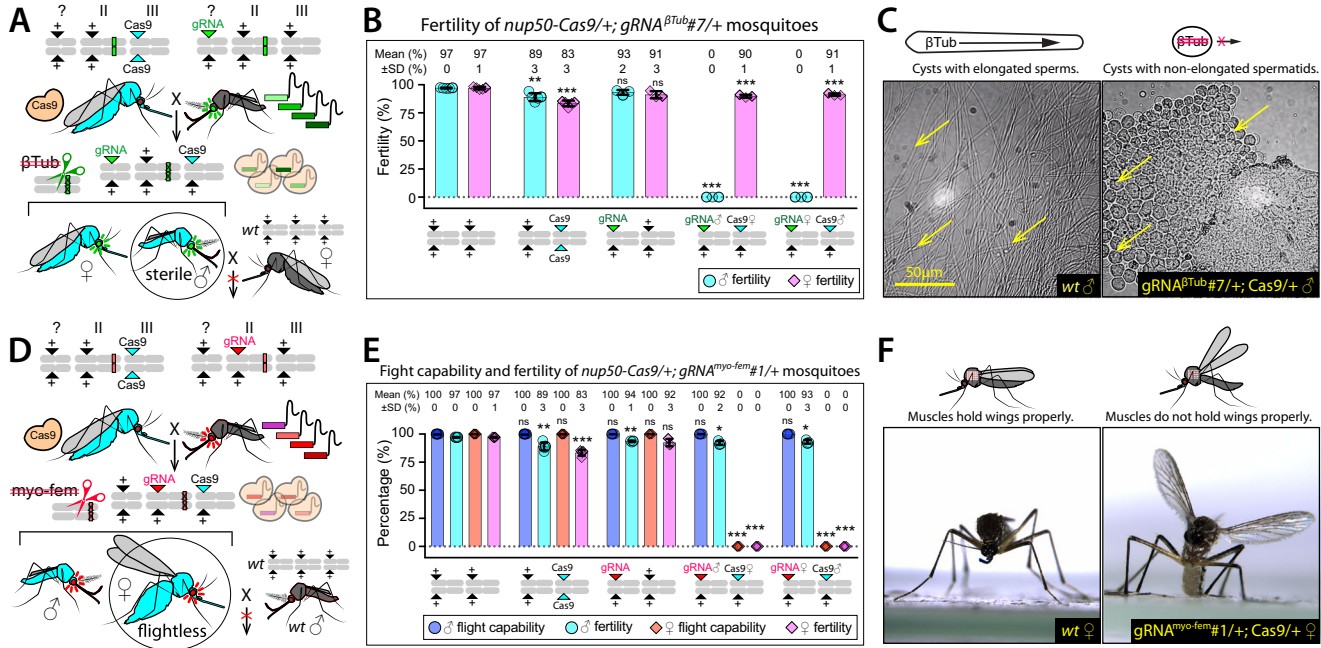

**Fig. 1 Validation of pgSIT target genes.** Cas9/gRNA-mediated disruption of *βTub* or *myo-fem* results in (**A–C**) male (♂) sterility or (**D, E**) female (♀) flightlessness, respectively. Schematics of genetic crosses to assess the efficiency of (**A**) *βTub* or (**D**) *myo-fem* disruption in the F₁ transheterozygous progeny. **B** Bar graph indicating the percent of fertile progeny for each of the various progeny genotypes using *gRNA^βTub#7* line. **C** Imaging of seminal fluid from wildtype (WT) and *gRNA^βTub#7/+;Cas9/+* mosquitoes, showing the difference in spermatid elongation caused by the disruption of *βTub* (Supplementary Video 1). Nonelongated spermatid phenotype was observed in each examined F₁ transheterozygous ♂. **E** Bar graph showing percent of fertile and flight-capable mosquitoes in each cross using *gRNA^myo-fem#1* line (Supplementary Videos 1–3). **F** Imaging showing the specific wing posture phenotype induced by the *myo-fem* disruption in ♀'s, but not in ♂'s, in which the resting wings were uplifted. Data from both paternal Cas9 crosses (Cas9♂ × gRNA♀) and maternal Cas9 crosses (Cas9♀ × gRNA♂) are shown (Supplementary Fig. 2, Supplementary Fig. 3, and Supplementary Table 3). Bar plots show means ± one standard deviation (SD) over at least three (n ≥ 3) biologically independent F₁ progeny groups, and mean and SD values rounded to a whole number. A two-sided F test was used to assess the variance equality. Statistical significance of mean differences was estimated using a two-sided Student's *t* test with unequal or equal variance. ($P \geq 0.05$ ^ns, $P < 0.05$*, $P < 0.01$**, and $P < 0.001$***). Source data are provided as a Source Data file.

period of time (2, 6, 12, 24, or 48 h) followed by WT ♂'s (48 h). Fertility was measured for up to five gonotrophic cycles. We found that prior exposure to *pgSIT^♂*'s ensured long-lasting reductions in ♀ fertility, spanning five gonotrophic cycles, with longer exposures (24 and 48 h) resulting in near-complete suppression of ♀ fertility (Fig. 2 and Supplementary Table 8).

**pgSIT-induced population suppression**. To assess whether *pgSIT* ♂'s could compete and suppress populations, we conducted discrete, multigenerational, population cage experiments by repeatedly releasing either pgSIT eggs or adult ♂'s each generation, using several introduction frequencies (pgSIT:WT - 1:1, 5:1, 10:1, 20:1, and 40:1) (Fig. 3 and Supplementary Table 16). To measure the efficacy of each generation, we counted the total number of laid and hatched eggs, and confirmed the absence of marker genes in hatched larvae indicating that released *pgSIT* ♂'s were indeed sterile. Adult releases at high-release thresholds (20:1, 40:1) eliminated all populations by generation 3, and at lower release thresholds (10:1), we observed gradual suppression followed by elimination by generation 6. For the egg releases, elimination was achieved by generation 6 for 4/6 populations at high-release thresholds (20:1, 40:1).

**Theoretical performance of pgSIT in a wild population**. To explore the potential for *pgSIT* ♂'s to suppress *Ae. aegypti* populations in the wild, we simulated releases of *pgSIT* eggs on the motu of Onetahi, Teti'aroa, French Polynesia (Fig. 4), a field site for releases of *Wolbachia*-infected ♂ mosquitoes, using the MGDrivE simulation framework[23]. Weekly releases of up to 400

*pgSIT* eggs (♂ and ♀) per WT adult (♂ and ♀) were simulated in each human structure over 10–24 weeks. The scale of these releases was chosen considering adult release ratios of 10:1 are common for sterile ♂ mosquito interventions[7] and ♀ *Ae. aegypti* produce >30 eggs per day in temperate climates[24]. We also assumed 25% reductions in ♂ mating competitiveness and adult lifespan for *pgSIT* ♂'s by default because, although *pgSIT* fitness effects were not apparent from laboratory experiments, they may become apparent in the field. Results from these simulations suggest that significant population suppression (>96%) is observed for a wide range of achievable release schemes, including 13 weekly releases of 120 or more *pgSIT* eggs per WT adult (Fig. 4 and Supplementary Video 7). Population elimination was common for larger yet achievable release schemes, including 18 weekly releases of 200 or more *pgSIT* eggs per wild adult, and 24 weekly releases of 100 or more *pgSIT* eggs per wild adult. Results also suggest a wider range of *pgSIT* fitness profiles (e.g., a 50% reduction in ♂ mating competitiveness and 25% adult life-span reduction) could lead to population elimination for these release schemes (Fig. 4).

## Discussion

While many technologies for halting the spread of deadly mosquito-borne pathogens exist, none are without significant drawbacks such that additional measures are needed. By disrupting essential genes throughout development, we demonstrate efficient production of short-lived, flightless *pgSIT^♀*'s and fit sterile *pgSIT^♂*'s. Importantly, when repeatedly released into caged populations, the *pgSIT^♂*'s competed with WT ♂'s thereby

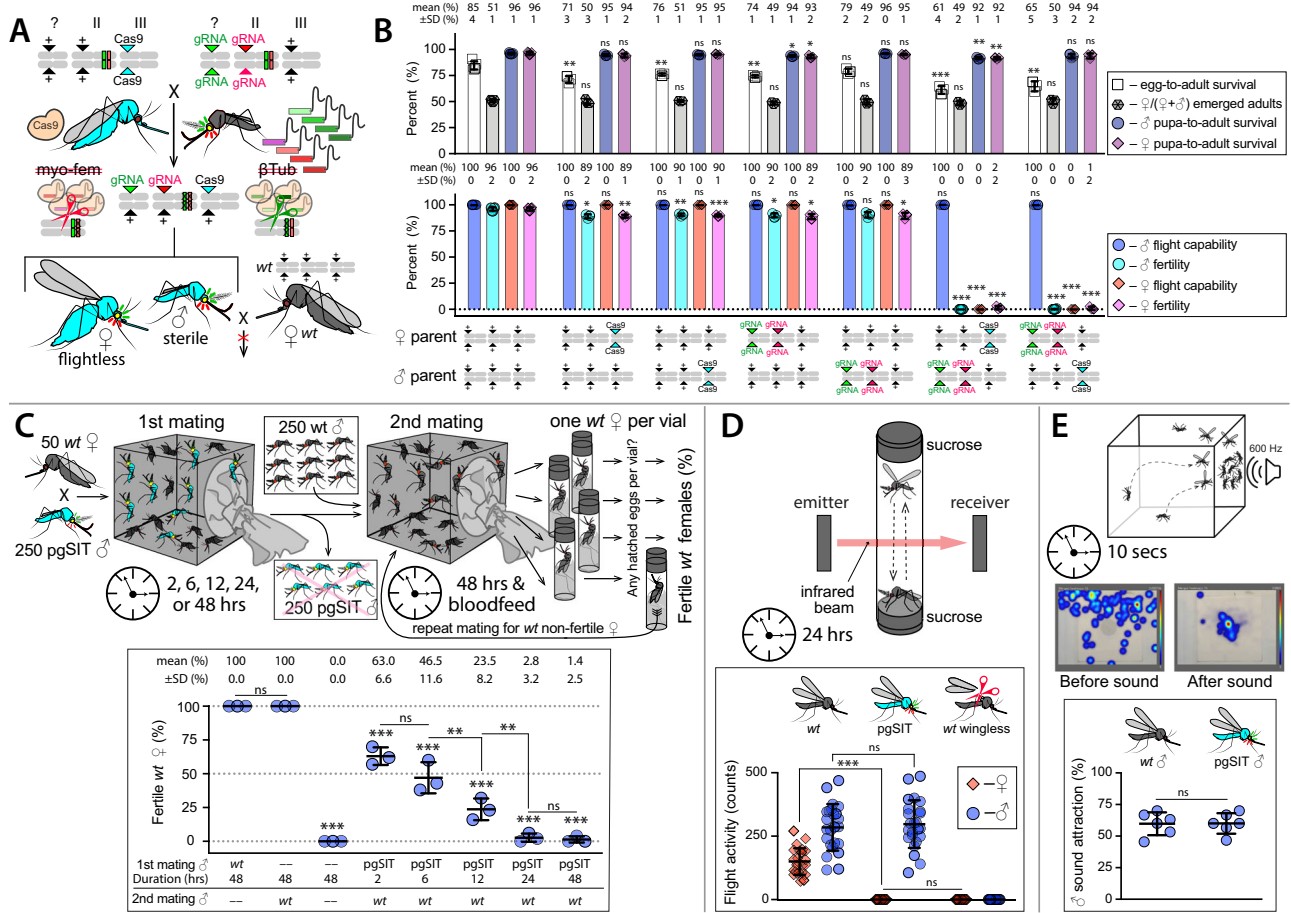

**Fig. 2 Genetic characterization of pgSIT. A** The pgSIT cross between double-homozygous gRNA ♂'s harboring both *gRNA*<sup>βTub</sup>#7 and *gRNA*<sup>myo-fem</sup>#1 (termed: *gRNA*<sup>βTub + myo-fem</sup>) and homozygous *Cas9* ♀'s. The pgSIT cross was initiated reciprocally to generate F₁ transheterozygous progeny carrying either maternal or paternal *Cas9* (Supplementary Fig. 4). **B** Bar graphs comparing the percentage of transheterozygous and heterozygous *Cas9* or gRNA progeny to those of WT (data can be found in Supplementary Table 4 and Supplementary Video 4) over at least three (*n* ≥ 3) biologically independent F₁ progeny groups. **C** Experimental setup to determine whether prior matings with *pgSIT*♂'s suppresses WT female (♀) fertility. WT ♀'s were cohabitated with *pgSIT*♂'s for 2, 6, 12, 24, or 48 h then WT ♀'s were transferred to a new cage along with WT ♂'s and mated for an additional 2 days. The ♀'s were then blood-fed and individually transferred to a vial. Eggs were collected and hatched for fertility determination. Following this, nonfertile ♀'s were then placed back into cages along WT ♂'s for another chance to produce progeny. This was repeated for up to five gonotrophic cycles, and the percentage of fertile ♀'s in each group of 50 ♀'s was plotted (Supplementary Table 8). The plot shows the fertility of three biologically independent groups of 50 WT ♀ (*n* = 3) for each experimental condition. **D** Flight activity of 24 individual mosquitoes (*n* = 24) was assessed for 24 h using a vertical *Drosophila* activity monitoring (DAM) system, which uses an infrared beam to record flight (Supplementary Table 6 and Supplementary Video 5). **E** To quantify the attractiveness of ♂'s to ♀'s for mating, we used a mating-behavior lure of a tone mimicking ♀ flight. A 10-s 600 Hz sine tone was applied on one side of the cage, and a number of mosquito ♂'s landing on the mesh around a speaker was scored. Heatmaps were generated using Noldus Ethovision XT (Supplementary Table 7 and Supplementary Video 6). The experiment was repeated six times (*n* = 6) using fresh groups of 30–40 ♂'s. Point plots (**C**–**E**) show biological replicates and means ± SDs. A two-sided F test was used to assess the variance equality. Statistical significance of mean differences was estimated using a two-sided Student's *t* test with unequal or equal variance. (*P* ≥ 0.05 <sup>ns</sup>, *P* < 0.05*, *P* < 0.01**, and *P* < 0.001***). Source data are provided as a Source Data File.

suppressing, and even eliminating, populations using release ratios that are achievable in the field[4,5,7]. Mathematical models suggest that population elimination could be accomplished in the field through sustained releases of ~100–200 or more *pgSIT* eggs per wild *Ae. aegypti* adult, even if fitness costs significantly exceed those measured in laboratory experiments.

For pgSIT to be realized in the wild, the two engineered strains will first need to be separately and continuously mass-reared in a facility, without contamination, and crossed to produce sterile ♂'s. While this can be viewed as rate-limiting[25], it offers stability, as the binary CRISPR system will remain inactive until crossed—thereby reducing the evolution of suppressors or mutations that could disrupt the system. In addition, each sorted ♀ can produce up to 450 eggs in her lifetime (~90 eggs per gonotrophic cycle)[26], which improves scalability. Moreover, once crossed, the resulting

progeny are essentially dead-ends (i.e., sterile ♂'s/flightless ♀'s), and flightless ♀'s hatched among high numbers of sterile *pgSIT*♂'s, should not contribute to the gene pool[27]. We demonstrate here that the technology is fully penetrant by screening >100 K individuals. Notwithstanding these results, it's possible that when this is scaled further to releasing millions, or even billions, there may be some error rate. What error rate is acceptable? What error rate will still enable population suppression? We do not have these answers, however, we can compare to Oxitec's OX531A RIDL system with the lethality trait penetrance being >95%[28,29]. This imperfect penetrance indicates that up to 5% of the OX531A offspring (♂'s and ♀'s) can survive and reproduce the following release. Despite this imperfect penetrance, OX531A transgenes did not persist/establish in wild populations[30], and overall, OX531A has been trialed with

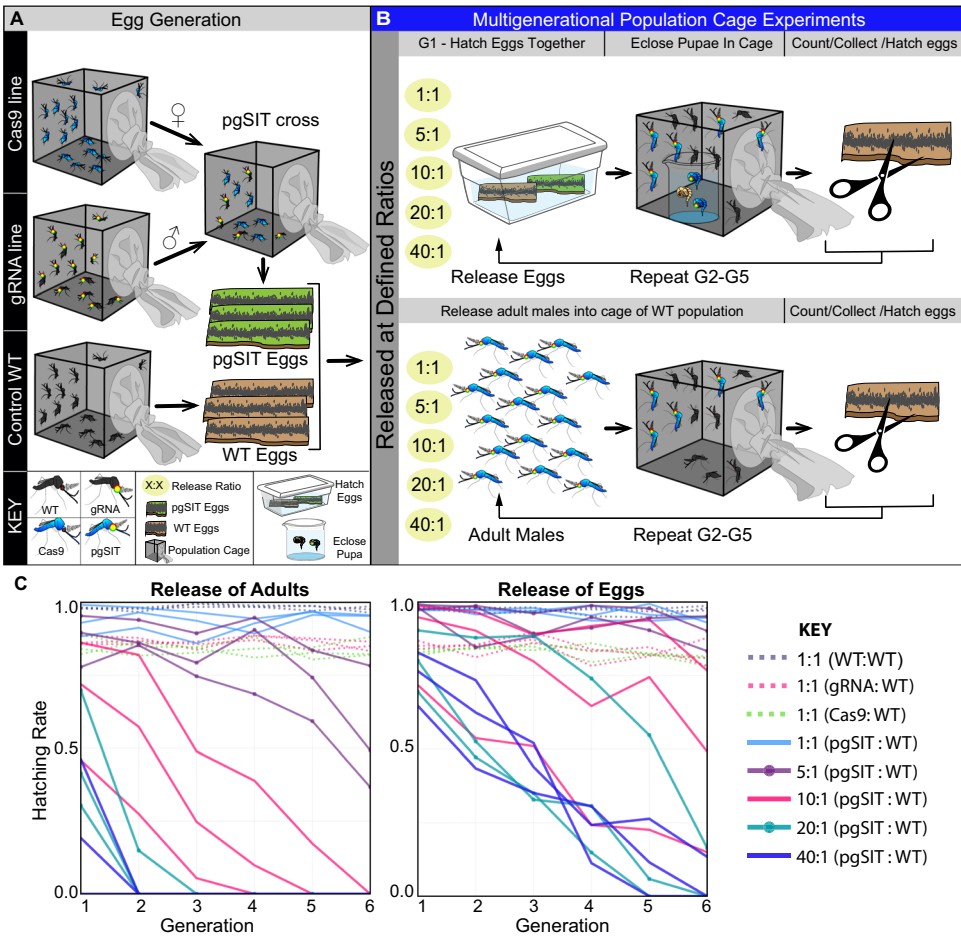

**Fig. 3 Multigenerational cage trials demonstrating efficient population suppression. A** To generate sufficient mosquito numbers, three lines were raised separately, including homozygous *Cas9*, double-homozygous *gRNA*$^{\beta Tub + myo\text{-}fem}$, and WT. To generate pgSIT progeny, virgin *Cas9* ♀'s were genetically crossed to *gRNA*$^{\beta Tub + myo\text{-}fem}$ ♂'s, and eggs were collected. **B** To perform multigenerational population cage trials of pgSIT, two strategies were employed: release of eggs (**B**, top panel); release of mature adults (**B**, bottom panel). For both strategies, multiple pgSIT:WT release ratios were tested, including: 1:1, 5:1, 10:1, 20:1, and 40:1. Each generation, total eggs were counted, and 100 eggs were selected randomly to seed the subsequent generation. The remaining eggs were hatched to measure hatching rates and score transgene markers. This procedure was repeated after each generation until each population was eliminated (Supplementary Table 16). **C** Multigenerational population cage data for each release threshold plotting the proportion of eggs hatched each generation. Source data are provided as a Source Data file.

outstanding success at suppressing populations in the wild[7]. Taken together, this indicates that, while perfection is the ultimate goal, it should not be necessary for obtaining approvals nor for achieving success in the field.

The pgSIT system developed here resulted in sterile ♂'s and flightless ♀'s. Importantly, if these flightless ♀'s were to mate to WT ♂'s, blood feed, and lay eggs, the transgene could be introduced into the local population. However, the likelihood is very low. First, *Ae. aegypti* ♂'s detect ♀'s via the sounds produced by ♀'s beating wings, and following a fast-paced, mid-air pursuit, they mate[31–34]. pgSIT ♀'s cannot fly nor hold wings properly (Fig. 1F), making them unable to produce these essential mating sounds and mate. Second, these flightless ♀'s move slowly and in a lethargic manner (Supplementary Video 2), with significantly reduced survival (Supplementary Fig. 5A, B and Supplementary Table 5). Taken together, the chance to find a blood source, blood feed, mate, and then find oviposition sites is extremely low. Third, the release process will continue for several rounds, even if the transgene is introduced into the local population, the population should be completely eliminated after several rounds of releases.

Finally, an egg-release device could be developed that would require the mosquitoes to fly out of the rearing container to enter the environment. This would basically eliminate the chances of a flightless ♀'s from entering the environment.

pgSIT offers an alternative approach to scalability that should help decrease costs and increase efficiency. For instance, the required genetic cross at scale can be initiated using existing robotic sex sorting devices (www.senecio-robotics.com) or an automated robotic system developed by Verily[5]. Upon sex sorting and crossing, the resulting *pgSIT* progeny can be distributed and released at any life stage, mitigating requirements for sex separation at field sites. This strategy will be especially effective for mosquitoes that diapause during the egg stage (e.g., *Aedes* species) because it will enable long-term egg accumulation. Eggs could be distributed to logistically spaced remote field sites where they can hatch, develop into adults, and fly out to compete with wild mosquitoes (Supplementary Fig. 11). These hatching containers could be engineered in such a way to require the adults to fly out thereby preventing the release of flightless ♀'s. This attractive feature should reduce the costs of developing multiple production

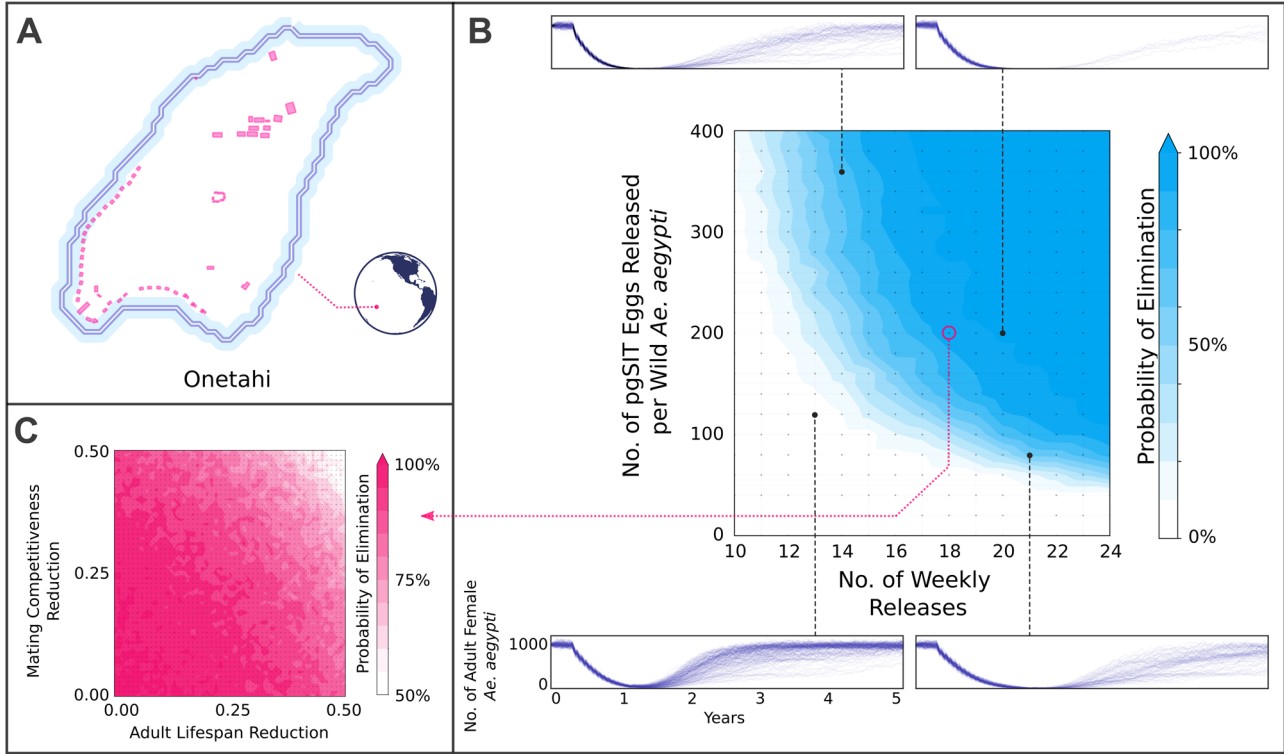

**Fig. 4 Model-predicted impact of releases of pgSIT eggs on *Ae. aegypti* population density and elimination. A** Releases were simulated on the motu of Onetahi (73.8 hectares), Teti 'aroa, French Polynesia, a field site for releases of *Wolbachia*-infected ♂ mosquitoes, using the MGDrivE simulation framework[23] and parameters described in Supplementary Table 17. Human structures are depicted and were modeled as having an equilibrium population of 16 adult *Ae. aegypti* each. **B** Weekly releases of up to 200 *pgSIT* eggs per WT *Ae. aegypti* were simulated in each human structure over 10–24 weeks. The *pgSIT* construct was conservatively assumed to decrease ♂ mating competitiveness by 25% and adult lifespan by 25%. Elimination probability was calculated as the percentage of 200 stochastic simulations that resulted in local *Ae. aegypti* elimination for each parameter set. Sample time series depicting WT ♀ *Ae. aegypti* population density is depicted above and below the heatmap. **C** Elimination probability (given 18 weekly releases of 200 *pgSIT* eggs per WT *Ae. aegypti*) is depicted for a range of *pgSIT* ♂ fitness profiles. Elimination is possible for a wide range of reductions in ♂ mating competitiveness (0–50%) and adult lifespan (0–50%) for an achievable release scheme. A Source Data file is provided.

facilities requiring on-site sex separation for the manual release of fragile adults. That said, to achieve adequate control in major urban settings, repeated releases will likely be required on a continuous basis and costs will need to be considered.

It should be noted that the releases of adult *pgSIT*♂'s unexpectedly resulted in faster population suppression as compared to egg releases in multigenerational population cage experiments. We believe this to result from the slightly reduced egg hatching rates of *pgSIT*♂'s and their delayed larva–pupa development time, which likely enabled the co-released WT ♂'s first access to WT ♀'s. While this could impact the discrete generation population cage experiments conducted here, it should not be problematic for suppressing continuous populations in the wild.

Finally, notwithstanding its inherently safe nature, pgSIT requires genetic modification, and regulatory use authorizations will need to be granted prior to implementation. While this could be viewed as a limitation[25], we do not expect obtaining such authorizations to be insurmountable. In fact, we envision pgSIT to be regulated in a similar manner to Oxitec's RIDL technology, which has been successfully deployed in many locations including the USA.

Overall, the inherent self-limiting nature of pgSIT, offers a controllable safe alternative to technologies that can persist and spread in the environment, such as gene drives[9]. Going forward, pgSIT may provide an efficient, safe, scalable, and environmentally friendly alternative next-generation technology for wild population control of mosquitoes resulting in wide-scale prevention of human disease transmission.

## Methods

**Mosquito rearing and maintenance**. *Ae. aegypti* mosquitoes were derived from the Liverpool strain (wild-type (WT)) previously used to generate the reference genome[35]. Mosquitoes were raised in incubators at 27.0 °C with 20–40% humidity and a 12-h light/dark cycle in cages (Bugdorm, 24.5 × 24.5 × 24.5 cm). Adults were provided 0.3 M aqueous sucrose ad libitum, and ♀'s were blood-fed on anesthetized mice for 2 consecutive days for ~15 min at a time. Oviposition substrates were provided ~3 days following the second blood meal. Eggs were collected and aged for ~4 days to allow for embryonic development, then were hatched in deionized $H_2O$ in a vacuum chamber. Roughly ~400 larvae were reared in plastic containers (Sterilite, 34.6 × 21 × 12.4 cm, USA) with ~3 liters of deionized $H_2O$, and fed fish food (TetraMin Tropical Flakes, Tetra Werke, Melle, Germany). For genetic crosses, to ensure ♀ virginity, pupae were separated and sexed under the microscope by sex-specific morphological differences in the genital lobe shape (at the end of the pupal abdominal segments just below the paddles) before being released to eclose in cages. These general rearing procedures were followed unless otherwise noted. Mosquitoes were examined, scored, and imaged using the Leica M165FC fluorescent stereomicroscope equipped with the Leica DMC2900 camera. For higher-resolution images, we used a Leica DM4B upright microscope equipped with a VIEW4K camera enabling time-lapse videos. Time-lapse videos of caged adult mosquitoes were taken with a mounted Canon EOS 5D Mark IV using a 24–105-mm image stabilizer ultrasonic lens.

**Guide RNA design and testing**. Two target genes were selected for gRNA design: *β-Tubulin 85D* (*βTub*, AAEL019894) and *myosin heavy chain* (*myo-fem*, AAEL005656). For each target gene, DNA sequences were first identified using reference genome assembly[35], and genomic target sites were validated using PCR amplification and Sanger sequencing (Supplementary Table 18 for primer sequences). Gene structures, transcripts, and exon–intron junction boundaries were carefully evaluated using comprehensive developmental transcriptome data[13,35] loaded into an internal genome browser. Target gRNA sequences were selected to be 20 bp (N20) in length, excluding the PAM (NGG)[36]. For in silico gRNA selection, we used either CHOPCHOP V3.0.0 or CRISPOR to minimize potential genomic off-target cleavage events. In total, we designed four gRNAs

targeting *βTub* and four gRNAs targeting *myo-fem* (Supplementary Table 18). To confirm gRNA activity in vivo, each gRNA was in vitro synthesized prior to construct design (Synthego, CA, USA). Then 100 ng/µl of gRNA was individually injected into 50 preblastoderm stage embryos (0.5–1 h old) derived from *Exu-Cas9* maternally depositing mothers, per previous embryo-injection protocols[18,21]. The surviving G0 progeny were pooled (2–5 individuals per pool), and genomic DNA was extracted using the DNeasy blood and tissue kit (Qiagen, Cat. No./ID: 69506) following the manufacturer's protocols. To molecularly characterize the induced mutations, target loci were PCR amplified from extracted genomic DNA, and the PCR products were gel-purified (Zymo Research, Zymoclean Gel DNA Recovery Kit, Cat. No./ID: D4007). The purified products were either sent directly for sequencing or subcloned (Invitrogen, TOPO-TA, Cat. No./ID: LS450641), wherein single colonies were selected and cultured in Laurel Broth (LB) with ampicillin before plasmid extraction (Zymo Research, Zyppy plasmid miniprep kit, Cat. No./ID: D4036) and Sanger sequencing. Mutated alleles were identified in silico by alignment with WT target sequences. All primers used for PCR and sequencing, including gRNA target sequences, are listed in Supplementary Table 18.

**Construct molecular design and assembly.** The Gibson enzymatic assembly method was used to engineer all constructs in this study[37]. To generate the *Nup50-Cas9* construct marked with *CFP*, OA-874PA (Addgene https://www.addgene.org/164846/), we used our previous plasmid for *Cas9* expression (Addgene plasmid https://www.addgene.org/100608/) as the backbone[21]. The fragments of *T2A-eGFP-P10-3′UTR* and *OpIE2-dsRed-SV40* were removed by cutting with restriction enzyme FseI. Then, the *P10-3′UTR* fragment was amplified from Addgene 100608 (https://www.addgene.org/100608/) with primers 874-P10 and 777B. Another fragment, *OpIE2-CFP-SV40*, was synthesized using gBlocks® Gene Fragment service (Integrated DNA Technologies, Coralville, Iowa). Both fragments were provided for the Gibson assembly into the cut backbone. We designed two constructs, OA-1067A1 (https://www.addgene.org/164847/) and OA-1067K (https://www.addgene.org/164848/), each carrying four different gRNAs targeting either β-Tubulin 85D (*βTub*, AAEL019894) or *myosin heavy chain* (*myo-fem*, AAEL005656) genes.

To engineer these plasmids, four intermediate plasmids, OA-1055A (*gRNAβTub1&2*), OA-1055B (*gRNAβTub3&4*), OA-1055W (*gRNAmyo-fem1&2*), and OA-1055X (*gRNAmyo-fem3&4*), each harboring two gRNAs, were generated by cutting a backbone plasmid OA-984 (https://www.addgene.org/120363/), which contains *piggyBac* elements and the *3xP3-tdTomato* transformation marker, with the restriction enzymes AvrII and AscI. Two gBlocks® Gene Fragments were then cloned in, each containing two gRNAs: one driven by *U6b* (AAEL017774) and one by *U6c* (AAEL017763) promoters[21]. To assemble the final plasmid OA-1067A1, an intermediate plasmid OA-1067A was generated by linearizing the plasmid OA-1055B with the restriction enzyme BglII and inserting in the fragment of *U6b-gRNAβTub1-U6c-gRNAβTub2* amplified with primers 1167.C1 and 1067.C2 from plasmid OA-1055A. Then, the fragment of *3xP3-tdTomato* was removed from plasmid OA-1067A using the restriction enzymes AscI and NotI and replaced with the *3xP3-eGFP* transformation marker amplified with primers 1067A1.C1 and 1067A1.C2 from the plasmid OA-961B (https://www.addgene.org/104967/). To assemble the final plasmid OA-1067K, OA-1055W was linearized with the restriction enzyme FseI, and the insertion of *U6b-gRNAmyo-fem3-U6c-gRNAmyo-fem4* was amplified with primers 1167.C5 and 1067.C6 from the plasmid OA-1055X. During each cloning step, single colonies were selected and cultured in LB medium with ampicillin, and then the plasmids were extracted (Zymo Research, Zyppy plasmid miniprep kit, Cat. No./ID: D4036) and Sanger sequenced. Final plasmids were maxi-prepped using (Zymo Research, ZymoPURE II Plasmid Maxiprep kit, Cat. No./ID: D4202) and Sanger sequenced. All primers are listed in Supplementary Table 18. Complete annotated plasmid sequences and plasmid DNA are available at Addgene.

**Generation of transgenic lines.** Transgenic lines were generated by microinjecting preblastoderm stage embryos (0.5–1 h old) with a mixture of the piggybac plasmid (200 ng/µl) and a transposase helper plasmid (*phsp-Pbac*, 200 ng/µl). Embryonic collection and microinjections were performed following established procedures[21]. After 4 days of development post-microinjection, G0 embryos were hatched in deionized H₂O in a vacuum chamber. Surviving G0 pupae were separated and sexed and divided into separate ♀ or ♂ cages (~20 cages total). The pupae enclosed inside these cages along with added WT ♂ pupae (added into the ♀ cages) or WT ♀ pupae (added into the ♂ cages) at 5:1 ratios (WT:G0). Several days post eclosion (~4–7), enabling sufficient time for development and mating, a blood meal was provided, and eggs were collected, aged, then hatched. The hatched larvae with positive fluorescent markers were individually isolated using a fluorescent stereomicroscope (Leica M165FC). To isolate separate insertion events, selected transformants were individually crossed to WT (5:1 ratios of WT:G1), and separate lines were established (Supplementary Table 2). These were subjected to many generations of backcrosses to WT to isolate single insertion events. Each of these individual gRNA lines (OA-1067A1: *gRNAβTub* and OA-1067K: *gRNAmyo-fem*) were maintained as mixtures of homozygotes and heterozygotes with periodic selective elimination of WTs. The Cas9 line (OA-874PA: *Nup50-Cas9*) was homozygosed by ~10 generations of single-pair sibling matings selecting individuals with the

brightest expressing transformation markers. Homozygosity was confirmed genetically by repeated test crosses to WT.

**Genetic testing of established lines.** To assess the activity of the transgenic lines generated, we performed a series of genetic crosses by releasing sexed pupae into cages. We first crossed gRNA lines (*gRNA* ♂ × WT ♀) to generate heterozygotes. We next reciprocally crossed heterozygous *gRNAβTub*/+ (lines #1–10) and the heterozygous *gRNAmyo-fem*/+ (lines #1–5), with homozygous *Nup50-Cas9*, hereafter *Cas9*, (1 ♂ × 10 ♀). To measure the fecundity, the resulting transheterozygous F₁ progeny (*gRNAβTub*/+; *Cas9*/+), or (*gRNAmyo-fem*/+; *Cas9*/+), were reciprocally crossed to WT's (50 ♂ × 50 ♀), keeping track of the grandparents' genotypes (Supplementary Figs. 2 and 3 and Supplementary Table 3). Control crosses of WT ♂ × WT ♀; WT ♂ × *Cas9* ♀; *Cas9* ♂ × WT ♀; *gRNA*/+ ♂ × *Cas9* ♀; *gRNA*/+ ♀ × *Cas9* ♂; *gRNA*/+ ♀ × WT ♂; and *gRNA*/+ ♂ × WT ♀ were also set up for comparisons (50 ♂ × 50 ♀). Adults were allowed to mate in the cage for 4–5 days, then blood meals were provided, and eggs were collected and hatched. The percentage of egg hatching (i.e., fertility) was estimated by dividing the total number of eggs laid by the total number of hatched eggs. Larvae-to-adult survival rates were calculated by dividing the total number of adults that emerged by the total number of larvae. Pupae–adult survival rates were calculated by dividing the number of adults by the total number of pupae. Flight capacity for each sex was calculated by dividing the total number that were flightless (observed by eye) by the total number of adult mosquitoes of that sex. Blood acquisition rates were calculated by dividing the number of blood-fed ♀'s by the total number of ♀'s. To investigate ♂ internal anatomical features, testes and ♂ accessory glands (*n* = 20) were dissected in 1% PBS buffer for imaging.

**Generation and characterization of gRNAβTub + myo-fem.** To generate *gRNAβTub + myo-fem*, we genetically crossed *gRNAβTub#7* (marked with *3xP3-GFP*) with *gRNAmyo-fem#1* (marked with *3xp3-tdTomato*). Resulting F₁ transheterozygotes *gRNAβTub#7*/+; *gRNAmyo-fem#1*/+ were subjected to multiple generations of single-pair sibling matings, carefully selecting individuals with the brightest expressing transformation markers, to generate a double-homozygous stock (termed: *gRNAβTub + myo-fem*). Zygosity was confirmed genetically by repeated test crosses to WT. To measure efficacy, we bidirectionally crossed *gRNAβTub + myo-fem* with Cas9 (50 ♂ × 50 ♀), generating F₁ transheterozygotes *gRNAβTub + myo-fem*/+; *Cas9*/+. Control crosses were also set up for comparisons: *gRNAβTub + myo-fem* ♂ × WT ♀; *gRNAβTub + myo-fem* ♂ × WT ♀; *Cas9* ♂ × *Cas9* ♀; *Cas9* ♂ × WT ♀; and *Cas9* ♀ × WT ♂; (50 ♂ × 50 ♀). To determine the fecundity and fertility, resulting transheterozygous F1's (~3 days old) were bidirectionally crossed to WT (50 ♂ × 50 ♀; ten replicates each). These were allowed to mate for ~2 days and then blood-fed. Afterward, eggs were collected for up to five consecutive gonotrophic cycles and hatched.

**Determination of transgene integration sites and copy number.** To determine the transgene insertion site(s) and copy number(s), we performed Oxford Nanopore DNA sequencing. We extracted genomic DNA using the Blood & Cell Culture DNA Midi Kit (Qiagen, Cat. No. 13343) from 20 adult transheterozygous *pgSIT♂*'s (3 days old) harboring all three transgenes (*Cas9*/+; *gRNAβTub#7*/+; *gRNAmyo-fem#1*/+), following the manufacturer's protocol. The sequencing library was prepared using the Oxford Nanopore SQK-LSK109 genomic library kit and sequenced on a single MinION flowcell (R9.4.1) for 72 h to generate an N50 read length for the set of 4088 bp. Basecalling was performed using ONT Guppy basecalling software version 4.4.1, generating 2.94 million reads above quality threshold Q≥7, which corresponds to 8.68 Gb of sequence data. To determine transgene copy number(s), reads were mapped to the AaegL5.0 reference genome[35] supplemented with transgene sequences (OA-1067A1: *gRNAβTub*; OA-1067K: *gRNAmyo-fem*; and OA-874PA: *Cas9*) using minimap2[38]. In total, 2,862,171 out of 2,936,275 reads (97.48%) were successfully mapped with a global genome-wide depth of coverage of 5.495. We calculated the mean coverage depth for all contigs in the genome (2310) and the three plasmids (OA-1067A1: *gRNAβTub*; OA-1067K: *gRNAmyo-fem*; and OA-874PA: *Cas9*) as well as normalized coverage (Supplementary Tables 9 and 10). Transgene coverage ranged from 5.1 to 7.6, and normalized coverage ranged from 0.93 to 1.38. As compared to the three chromosomes, the coverages are consistent with the transgenes present at a single copy (Supplementary Fig. 7).

To identify transgene insertion sites, we inspected reads that aligned to the transgenes in the Interactive Genomics Viewer (IGV) browser. The reads extending beyond the boundaries of the transgenes were then analyzed to determine mapping sites within the genome. For OA-874PA, one read spanned the whole transgene (~11.5 kb) and extended 4 and 3.5 kb on both sides. The extending portions mapped to both sides of the position on NC_035109.1:33,210,105 (chromosome 3), with the nearest gene being AAEL023567, which is ~5 kb away. For OA-1067K, one read covered ~7 kb of the transgene extending ~10 kb off the 3′ end, 9 kb of which map to the NC_035108.1:287,686-296,810 region (chromosome 2). A few other shorter reads map to the same location. The site is located in the intron of AAEL005206, which is a capon-like protein, and based on the RNA-seq data, its expression does not appear to be affected in pgSIT animals. For OA-1067A1, the nanopore sequencing was unable to resolve the insertion site, presumably due to its

insertion in one of the remaining gaps in the genome. Finally, using nanopore data, we confirmed genomic deletions in both pgSIT target genes—see AEL019894 and AAEL005656 as expected (Supplementary Figs. 8 and 9). The nanopore sequencing data have been deposited to the NCBI sequence read archive (NCBI-SRA, PRJNA699282).

**Transcriptional profiling and expression analysis**. To quantify target gene reduction and expression from transgenes as well as to assess global expression patterns, we performed Illumina RNA sequencing. We extracted total RNA using miRNeasy Mini Kit (Qiagen, Cat. No. 217004) from ten sexed pupae: WT ♀, WT♂, transheterozygous $pgSIT^{♂}$'s, and $pgSIT^{♀}$ harboring all three transgenes $Cas9/+$; $gRNA^{βTub#7}/+$; $gRNA^{myo-fem#1}/+$ with each genotype in biological triplicate (12 samples total), following the manufacturer's protocol. DNase treatment was conducted using DNase I, RNase-free (ThermoFisher Scientific, Cat. No. EN0521), following total RNA extraction. RNA integrity was assessed using the RNA 6000 Pico Kit for Bioanalyzer (Agilent Technologies #5067-1513), and mRNA was isolated from ~1 µg of the total RNA using NEBNext Poly(A) mRNA Magnetic Isolation Module (NEB #E7490). RNA-seq libraries were constructed using the NEBNext Ultra II RNA Library Prep Kit for Illumina (NEB #E7770) following the manufacturer's protocols. Briefly, mRNA was fragmented to an average size of 200 nt by incubating at 94 °C for 15 min in the first strand buffer. cDNA was then synthesized using random primers and ProtoScript II Reverse Transcriptase followed by second-strand synthesis using NEB Second Strand Synthesis Enzyme Mix. The resulting DNA fragments were end-repaired, dA-tailed, and ligated to NEBNext hairpin adapters (NEB #E7335). Following ligation, adapters were converted to the "Y" shape by treating with USER enzyme, and DNA fragments were size-selected using Agencourt AMPure XP beads (Beckman Coulter #A63880) to generate fragment sizes between 250 and 350 bp. Adapter-ligated DNA was PCR amplified followed by AMPure XP bead clean up. Libraries were quantified using a Qubit dsDNA HS kit (ThermoFisher Scientific #Q32854), and the size distribution was confirmed using a High Sensitivity DNA Kit for Bioanalyzer (Agilent Technologies #5067- 4626). Libraries were sequenced on an Illumina HiSeq2500 in single-read mode with the read length of 50 nt and sequencing depth of 20 million reads per library. Base calls were performed with RTA 1.18.64 followed by conversion to FASTQ with bcl2fastq 1.8.4. The reads were mapped to the AaegL5.0 (GCF_002204515.2) genome supplemented with OA-874PA, OA-1067A1, and OA-1067K sequences using STAR. On average, ~97.5% of the reads were mapped (Supplementary Table 11). Gene expression was then quantified using feature-Counts against the annotation release 101 GTF (https://www.ncbi.nlm.nih.gov/assembly/GCF_002204515.2) downloaded from NCBI. TPM values were calculated from counts produced by featureCounts and combined (Supplementary Table 12).

PCA and hierarchical clustering of the data show that the samples generally behaved as expected in clustering by sex and genotype (Supplementary Fig. 10). DESeq2 was then used to perform differential expression analyses between pgSIT vs WT samples within each sex (Supplementary Fig. 10 and Supplementary Tables 13 and 14), and a two-factor design consistently showed what changed in response to the genotype in both sexes (Supplementary Table 15). In a comparison between $pgSIT^{♀}$ and WT ♀, 660 genes were upregulated in $pgSIT^{♀}$ and 392 were downregulated at an adjusted $P$ value <0.05. The target gene, AAEL005656, was significantly downregulated in $pgSIT^{♀}$ (Supplementary Fig. 10C). In a comparison between $pgSIT^{♂}$'s and WT♂ (Supplementary Table 13), 2067 genes were upregulated in $pgSIT^{♂}$'s and 2722 were downregulated at an adjusted $P$ value <0.05. The target gene, AEL019894, was strongly downregulated in $pgSIT^{♂}$ (Supplementary Fig. 10D). It is important to note here that the CRISPR/Cas9 pgSIT system disrupts the DNA (not the RNA) so transcription is expected to occur; however, the transcripts produced will encode mutations and should be degraded by nonsense-mediated mRNA decay (NMD) mechanisms. Indeed, these mutant RNA's can be observed in the IGV (Supplementary Figs. 8 and 9). In the two-factor comparison, 1447 genes were upregulated in pgSIT and 2563 were downregulated at an adjusted $P$ value <0.05 (Supplementary Fig. 10E). For each DESeq2 comparison, gene ontology enrichments were performed on significantly differentially expressed genes, and these are provided as tabs in the corresponding tables (Supplementary Tables 13–15). All Illumina RNA sequencing data have been deposited to the NCBI-SRA, PRJNA699282.

**Amplicon sequencing of target loci**. To sample a variety of molecular changes at the gRNA target sites (*myo-fem* and *βTub*), we used the Amplicon-EZ service by Genewiz® and followed the Genewiz® guidelines for sample preparation. Genomic DNA from 50 WT and 50 pgSIT sexed pupae (25♀ + 25♂) were extracted separately using DNeasy Blood and Tissue Kit (Qiagen, Cat. No./ID: 69506) following the manufacturer's protocols. Primers with Illumina adapters (Supplementary Table 18) were used to PCR amplify the genomic DNA. PCR products were purified using the Zymoclean Gel DNA Recovery Kit (Zymo Research, Cat. No./ID: D4007). Roughly 50,000 one-directional reads were generated by Genewiz® and uploaded to Galaxy.org for analysis. Quality control for the reads was performed using FASTQC. Sequence data were then paired and aligned against the *myo-fem* or *βTub* sequence using Map with BWA-MEM under "Simple Illumina mode". Sequence variants were detected using FreeBayes, with parameter selection level set to "simple diploid calling." The amplicon sequencing data has been provided as Supplementary File S1.

**Prior mating with $pgSIT^{♂}$'s suppress ♀ fertility**. To determine whether prior matings with $pgSIT^{♂}$ could reduce ♀ fertility, we initiated 15 cages each consisting of 250 mature (4–5 days old) $pgSIT^{♂}$ combined with 50 mature (4–5 days old) WT virgin ♀. We allowed the $pgSIT^{♂}$'s to mate with these ♀'s for a limited period of time (including 2, 6, 12, 24, and 48 h; all experiments begin at 9:00 am PST, three replicate cages each). Cages were shaken every 3 min for the first half an hour to increase mating opportunities. Following these time periods, all ♀'s were removed and transferred to new cages along with 250 WT mature ♂'s, cages were again shaken every 3 min for the first half-hour to increase mating opportunities and left to mate for an additional 2 days. The ♀'s were then blood-fed, and each blood-fed ♀ was individually transferred to a single narrow Polystyrene vial (Genesee Scientific, Cat. No. 32-116), and eggs were collected and hatched for fertility determination. Following this, nonfertile ♀'s were then placed back into cages along with the original WT ♂'s, plus an additional 50 mature WT ♂'s, for another chance to produce progeny. This was repeated for up to five gonotrophic cycles. As controls, cages with 250 WT ♂'s and 50 WT ♀'s, or 50 unmated blood-fed WT ♀'s with no ♂'s added, or 50 unmated blood-fed WT ♀'s with 250 WT ♂ adults were also set up (Supplementary Table 8).

**Life table parameters**. Life table parameters were assessed by comparing WT, homozygous $gRNA^{βTub + myo-fem}$, homozygous $Cas9$, and transheterozygous pgSIT ($gRNA^{βTub + myo-fem}/+$; $Cas9/+$) generated with $Cas9$ inherited from either the mother (maternal $Cas9$) or father (paternal $Cas9$). Larva/pupae development times were recorded as the number of days from hatched larvae to pupae and then to adults. One hundred larvae from each line were placed in separate larval rearing containers (Sterilite, 34.6 × 21 × 12.4 cm, USA), each with 3 liters of deionized water, and fed once a day. Larvae were counted twice daily until pupation, and then the date of pupation and emergence were recorded. Larval to pupae development time was calculated for each sex. Pupae were transferred to plastic cups (Karat, C-KC9) with 100 ml of water, and survivors were recorded until adulthood.

For measuring ♂ /♀ longevity, we tested the variation in ♂ and ♀ longevity among different lines using two methods: (i) released along with WT of the opposite sex or (ii) without WT of the opposite sex. (i) One hundred WT, homozygous $gRNA^{βTub + myo-fem}$, homozygous $Cas9$ newly eclosed adult mosquitoes (fifty ♂'s and fifty ♀'s) were maintained in a cage; fifty newly eclosed pgSIT ♂'s (maternal $Cas9$) and fifty newly eclosed pgSIT ♂'s (paternal $Cas9$) were caged with fifty newly eclosed WT ♀'s; and finally, fifty newly eclosed pgSIT ♀'s (maternal $Cas9$) and fifty newly eclosed pgSIT ♀'s (paternal $Cas9$) were caged with fifty newly eclosed WT ♂'s. (ii) Fifty ♂'s or ♀'s from each line were released into a cage separately without the opposite sex. Adults were provided with 10% sucrose and monitored daily for survival until all mosquitoes had died (three replicates).

For measuring ♀ fecundity and fertility, ♀'s ($n = 50$) and ♂'s ($n = 50$) 3 days post emergence raised under the same standardized larval conditions were placed into a cage and allowed to mate for 2 days. ♀ mosquitoes were blood-fed until fully engorged and were individually transferred into plastic vials with oviposition substrate. Eggs were stored in the insectary for 4 days to allow full embryonic development and then were hatched in a vacuum chamber. Fecundity was calculated as the number of eggs laid per ♀, and fertility was calculated as the percentage of eggs hatched per ♀.

♂ mating capacity (how many ♀'s can be mated by one mature ♂) was measured as follows. Fifteen mature WT ♀'s were caged with 1 mature ♂ of each genotype for 24 hours (1♂:15♀ ratio). After 24 h, the single ♂ was removed from all cages. Two days after the single ♂ was removed, 75 WT ♂'s were added to each cage that previously had a pgSIT ♂ (5♂:1♀ ratio). Blood meals were provided, and each blood-fed ♀ was individually transferred to a single vial for egg collection. The fecundity and fertility of each ♀ was determined. The mating capacity was calculated as the total number of ♀'s-total number of fertile ♀'s. The mating capacity of WT, homozygous $gRNA^{βTub + myo-fem}$, and homozygous $Cas9$ ♂ was equal to the number of fertile ♀'s.

**Flight activity quantification**. Mosquitoes were reared at 28 °C, 80% relative humidity under a 12:12 h light:dark regime, and measurements of flight activity were performed using a *Drosophila* Activity Monitoring (DAM) System (TriKinetics, LAM25) and DMASystem3 software (TriKinetics) using large tubes designed for mosquitoes (TriKinetics, PGT 25 × 125 mm Pyrex Glass). Individual 4–7-day-old, non-blood-fed virgin ♀ and non-mated ♂ mosquitoes were introduced into the monitoring tubes, which contained 10% sucrose (Sigma, Cat. No. S0389) at both ends of the tube as the food source. The DAM System was positioned vertically during the assays. Flight activity was measured over a period of 24 h by automatically calculating the number of times that mosquitoes passed through the infrared beam in the center of the tubes. The walls of the monitoring tubes were coated with Sigmacote (Sigma, Cat. No. SL2) to inhibit mosquitoes from walking upward. For preparing the wingless mosquitoes, the animals were anesthetized on ice, and the wings were removed using Vannas Scissor (World Precision Instruments, Cat. No. 14003). The wingless mosquitoes were allowed to recover for 12 h before recording. Mosquitoes were manually checked after flight activity recording to ensure survival. Data acquisition was performed using the DAM System (TriKinetics) (Fig. 2D, Supplementary Video 5, and Supplementary Table 6).

**Sound attraction assay**. The sound attraction assay was performed in a chamber with a temperature of 28 °C and humidity of 80%. Seven-day-old ♂'s were sex separated after the pupae stage. The day before testing, 30–40 ♂'s were transferred by mouth aspiration to a 15-cm$^3$ mesh cage with a 10% sucrose bottle. ♂ mosquitoes were allowed to recover in the cage under a 12 h:12 h light:dark regime for 24 h. For each trial, a 10-s 600 Hz sine tone was applied on one side of the cage as a mating-behavior lure, mimicking ♀ flight tones. The number of mosquitoes landing on the mesh area around the speaker box (10 cm$^2$) was quantified at 5-s intervals throughout the stimulus. The average percent of mosquitoes landing around the speaker area out of the total cage post-sound presentation was calculated (Fig. 2E, Supplementary Video 6, and Supplementary Table 7). Heatmaps were generated using Noldus Ethovision XT 16.

**Multigenerational population cage trials**. To perform multigenerational population cage trials, two strategies were employed: (i) release of eggs; (ii) release of mature adults (Fig. 3 and Supplementary Table 16). Cage trials were carried out using discrete non-overlapping generations. For the first release of eggs strategy (i), pgSIT eggs and WT eggs were hatched together using the following ratios (pgSIT: WT) of 1:1 (100:100), 5:1 (500:100), 10:1 (1000:100), 20:1 (2000:100), and 40:1 (4000:100), and three biological replicates for each ratio (15 cages total). All eggs were hatched simultaneously, then separated into multiple plastic containers (Sterilite, 34.6 × 21 × 12.4 cm, USA). Roughly 400 larvae were reared in each container using standard conditions with 3 liters of deionized water and were allowed to develop into pupae. Pupae were placed in plastic cups (Karat, C-KC9) with ~100 ml of water (~150 pupae per cup) and transferred to large cages (BugDorm, 60 × 60 × 60 cm) to eclose. All adults were allowed to mate for ~5–7 days. ♀'s were blood-fed, and the eggs were collected. Eggs were counted and stored for ~4 days to allow full embryonic development, then 100 eggs were selected randomly and mixed with pgSIT eggs with ratios (pgSIT: WT) of 1:1 (100:100), 5:1 (500:100), 10:1 (1000:100), 20:1 (2000:100), and 40:1 (4000:100) to seed for the following generation, and this procedure continued for all subsequent generations. The remaining eggs were hatched to measure hatching rates and to screen for the possible presence of transformation markers. The hatching rate was estimated by dividing the number of hatched eggs by the total number of eggs.

For the release of mature adults strategy (ii), 3–4-days-old mature WT adult ♂'s were released along with mature (3–4 days old) pgSIT adult ♂'s at release ratios (pgSIT: WT): 1:1 (50:50), 5:1 (250:50), 10:1 (500:50), 20:1 (1000:50), and 40:1 (2000:50), with three biological replicates for each release ratio (15 cages total). One hour later, 50 mature (3–4 days old) WT adult ♀'s were released into each cage. All adults were allowed to mate for 2 days. ♀'s were then blood-fed and eggs were collected. Eggs were counted and stored for 4 days to allow full embryonic development. Then, 100 eggs were randomly selected, hatched, and reared to the pupal stage, and the pupae were separated into ♂ and ♀ groups and transferred to separate cages. Three days post eclosion, ratios (pgSIT: WT) of 50 (1:1), 250 (5:1), 500 (10:1), 1000 (20:1), and 2000 (40:1) age-matched pgSIT mature ♂ adults were caged with these mature ♂'s from 100 selected eggs. One hour later, mature ♀'s from 100 selected eggs were transferred into each cage. All adults were allowed to mate for 2 days. ♀'s were blood-fed, and eggs were collected. Eggs were counted and stored for 4 days to allow full embryonic development. The remaining eggs were hatched to measure hatching rates and to screen for the possible presence of transformation markers. The hatching rate was estimated by dividing the number of hatched eggs by the total number of eggs. This procedure continued for all subsequent generations.

**Statistical analysis**. Statistical analysis was performed in JMP8.0.2 by SAS Institute Inc and Prism9 for macOS by GraphPad Software, LLC. At least three biological replicates were used to generate statistical means for comparisons. $P$ values were calculated for a two-sided Student's $t$ test with equal or unequal variance. A two-sided F test was used to assess the variance equality. The departure significance for survival curves was assessed with the Log-rank (Mantel–Cox) and Gehan–Breslow–Wilcoxon texts. Multiple comparisons were corrected by the Bonferroni method. All plots were constructed using Prism 9.1 for macOS by GraphPad Software, LLC.

**Mathematical modeling**. To model the expected performance of pgSIT at suppressing and eliminating local *Ae. aegypti* populations, we used the MGDrivE simulation framework[23]. While MGDrivE was designed for gene drives, thanks to the modular design, it is quite flexible at handling any inheritance pattern describable using Punnett squares (see ref. [39]). This framework models the egg, larval, pupal, and adult mosquito life stages with overlapping generations, larval mortality increasing with larval density, and a mating structure in which ♀'s retain the genetic material of the adult ♂ with whom they mate for the duration of their adult lifespan. The inheritance pattern of the *pgSIT* system was modeled within the inheritance module of MGDrivE, along with impacts on adult lifespan, ♂ mating competitiveness, and pupatory success. We distributed *Ae. aegypti* populations according to human structures sourced from OpenStreetMap on the basis that *Ae. aegypti* is anthropophilic. Each human structure was assumed to have an equilibrium population of 16 adult *Ae. aegypti*, producing an equilibrium motu population of 992 based on estimates from field collections on the motu[40]. We

implemented the stochastic version of the MGDrivE framework to capture random effects at low population sizes and the potential for population elimination. Weekly releases of up to 400 pgSIT eggs per wild adult mosquito (♂ and ♀) were simulated in all human structures of Onetahi over a period of 10–24 weeks. Egg-release size was calculated following the precedent in Kandul et al.[11]: density-independent mortality rates for juvenile life stages were calculated for consistency with the population growth rate in the absence of density-dependent mortality, while density-dependent mortality was applied to the larval stage following Deredec et al.[41], and both rates were used to calculate the expected number of eggs required to match a 10:1 adult (♂ and ♀) release ratio. 200 repetitions were carried out for each parameter set, and mosquito genotype trajectories, along with the proportion of simulations that led to local population elimination, were recorded. Complete model and intervention parameters are listed in Supplementary Table 17.

## Ethical conduct of research

All animals were handled in accordance with the Guide for the Care and Use of Laboratory Animals as recommended by the National Institutes of Health and approved by the UCSD Institutional Animal Care and Use Committee (IACUC, Animal Use Protocol #S17187) and UCSD Biological Use Authorization (BUA #R2401).

**Reporting summary**. Further information on research design is available in the Nature Research Reporting Summary linked to this article.

## Data availability

Complete sequence maps assembled in the study are deposited at Addgene.org (#164846 - #164848) and available for distribution (Supplementary Table 18). All Illumina and Nanopore sequencing data have been deposited to the NCBI-SRA https://www.ncbi.nlm.nih.gov/sra/?term=PRJNA699282. All data used to generate figures are provided in the Supplementary Materials/Source Data files (Supplementary Tables 1–18). Generated *Aedes* transgenic lines are available upon request to O.S.A. Source data are provided with this paper.

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

## Acknowledgements

We thank Judy Ishikawa for helping with mosquito husbandry. This work was supported by funding from a DARPA Safe Genes Program Grant (HR0011-17-2-0047) and an NIH award (R01AI151004) awarded to O.S.A. support to CM from the "U.S. Army Research Office and accomplished under cooperative agreement W911NF-19-2-0026 for the Institute for Collaborative Biotechnologies", support to J.M.M. from the Innovative Genomics Institute, and an NIH award (R56-AI153334). The views, opinions, and/or findings expressed are those of the authors and should not be interpreted as representing the official views or policies of the U.S. government.

## Author contributions

O.S.A. and M.L. conceptualized and designed experiments; T.Y., J.L., L.A., and S.G. performed molecular analyses; M.L., J.R.E., T.W., H.L., and M.B. performed genetic experiments; Y.Z., Y.W., N.D., J.C., and C.M. performed behavioral experiments; J.B., H.M.S.C., and J.M.M. performed mathematical modeling; I.A. performed bioinformatics: O.S.A., M.L., N.P.K., and R.R. analyzed and compiled the data. All authors contributed to writing and approved the final manuscript.

## Competing interests

O.S.A. is a founder of Agragene, Inc. and has an equity interest. The terms of this arrangement have been reviewed and approved by the University of California, San Diego in accordance with its conflict of interest policies. All remaining authors declare no competing interests.
