## [Transparent Peer Review file · Nature Communications]

Peer Review Information

Manuscript title: Suppressing Mosquito Populations with Precision Guided Sterile Males

Corresponding author name(s): Omar Akbari

Reviewer comments & decisions:

Reviewer comments, first version:

Reviewer #1 (Remarks to the Author: Overall significance):

High significance. **More details in my attached summary.**

Reviewer #1 (Remarks to the Author: Impact):

High impact. **More details in my attached summary.**

Reviewer #1 (Remarks to the Author: Strength of the claims):

Convincing and well-supported by data. The modeling work needs more information, as I mention in my one major critique **in the attached summary**.

Reviewer #1 (Remarks to the Author: Reproducibility):

I did not see any problem with the reproducibility or statistics.

Reviewer #2 (Remarks to the Author: Overall significance):

The paper brings a very interesting approach on how to obtain completely sterile males and removal of 100% females.

The title is misleading for something not reflected in the manuscript results. Half of the results are missing in the abstract section, with no mention about the flightless mosquitoes nor the combination of

the two strategies.

For the figures, all legends are incomplete. And the figures are extremely hard to comprehend and pollute. For the supplementary material should complement the manuscript and bring useful information, but the way is presented is that every single possible way to present something that sound as a result was included. And their majority could have been included in the text. This reflects a confusing thinking about the supplementary material selection. The main figures are so crowded that part of them should be the supplementary material, replacing most of the current ones.

The manuscripts attempt to bring a fulfillment and grandiosity intimidating the readers and leading them to a mental confusion and accepting everything that was presented.

Lines 117-120: What is the potential reason for the other non-selected strains to have variable fertility levels and even full fertility? Same for flying females of non-selected pgSIT strain.

Reviewer #2 (Remarks to the Author: Impact):

Nothing to declare

Reviewer #2 (Remarks to the Author: Strength of the claims):

Lines 125-128: Why authors were so confident that they would have superior results and “robust” transgene expression? Specially, when other strains obtained with the same construction are fertile (Figure S2) and some females even though they were flightless, some were still able to blood feed (Figure 3).

Lines 132-134: Linked to the previous comment and the speculative sentence on these lines, the potential application of egg release for population suppression, and given the unrealistic possibility that these females may find conditions to mate, blood feed and lay eggs, how would be the potential impact of these offspring? Since regulatory bodies may find this information quite alarming.

Lines 137-142: It's known that is not only time but also the moment that the coitus is interrupted so a second male can contribute with sperm to the offspring (DOI: 10.1371/journal.pone.0193164). By number used at the beginning of the experiment (which is also not clear the methods) the male sexual harassment over females may be very high, and within the 2-6 hours, males keep removing each other during coitus or several males trying to mate at same time with a flying female, causing the tendency of females not to move/fly to avoid coitus. The other possibility is the mating vigor (which is not the mating capacity as shown) and males need longer periods to mate.

Lines 151-154: If as stated in the manuscript, no significant impact on the fitness cost was found, however comparing these result with a study using ionizing gamma radiation on male mosquitoes, in both studies the population suppression took 6 generations to collapse using the same sterile : fertile ratio (DOI: 10.1098/rstb.2019.0808), and using eggs for releases indicates a higher impact on fitness by the use of much higher ratios (20 : 1 or 40 : 1) for the same number of generations to reach population

suppression.

Lines 164-167: As there were no field releases, why 25% reduction in male competitiveness was chosen? Another indication that the pgSIT males have lower fitness, is the comparison with a real field release in Brazil performing a mark-release-recapture experiment using drones and gamma irradiated males, in which within couple of releases (in the same period) they manage to observe more than 50% reduction of viable eggs in the field compared with the control site (DOI: 10.1126/scirobotics.aba6251).

Line 181: I am skeptical to call the strain 100% fit.

Lines 181-183: The cage suppression experiment is based on a discrete and multigenerational process, which is quite artificial and not realistic. A similar experiment established a wild population in the semi-field cages, before starting releasing their “sterile” males. And because they also followed control cages during the study period, the replicates showed it is clear the impact and the low variability among the same sample group (DOI: 10.1038/s41586-019-1407-9).

Lines 183-186: Excessive negative fitness cost can be a problem during operational releases, even in a small trial. The transgenic strain used in Brazil (OX513A) was known to have negative fitness cost (DOI: 10.1371/journal.pone.0020699 and DOI: 10.1371/journal.pone.0026086). And the outcome of this was seen overlapping the studies using this same strain for mass production and population suppression (DOI: 10.3791/3579 and DOI: 10.1371/journal.pntd.0003864). Although the authors claim to have achieved population suppression around 70 to more than 90%, the released area was quite small (5.5 ha with around 500 residences) and required a production of 0.5 million males/week to reach this result.

Line 188: It is written “realized” but do you mean “released”?

Line 192: females produce 450 eggs throughout their life, this is a very high assumption since this would be around 90 eggs/ gonotrophic cycle, which may not be realistic depending on strain, blood quality, oviposition site and other factors.

Lines 198-199: Although field models show promising results using pgSIT, the scalability was not modeled and tested under realistic mass rearing facility to attend areas of pilot trials that are slightly more challenged than Onetahi island.

Line 200: Senecio’s technology so far is not available for operational trials, and Debug haven’t tried transferring their technology to other operational trials with bigger populations and the maximum mentioned was less than 3 million males/week, which depending on the trial site is not enough to suppress the population.

Lines 204-207: Egg releases sound not as a promising technique since there will be no control on the male quality produced. And the dissemination of these eggs might be as laborious as adult production and release, unless the target is only to produce a product and not vector control. How would be the dynamics of deploying dissemination stations and how would be the entomological surveillance of these areas?

Line 201-214: However, the proposed model also showed that egg release was less effective as adult releases, meaning that the suppression delay will persist.

Figure 4: According to some quick calculations using data from the manuscript only for Onetahi, which seems to be a quite isolated area and without a real mosquito burden and disease transmission, using

the pgSIT it would be necessary to obtain 0.8 million eggs of the strain to be released every week. And a facility to keep this production that depends on two strains to obtain, will definitely be a complex process since they must be independent facilities, multiplying by two the area, team, equipment all consumables in order to produce for a quite small trial.

Reviewer #2 (Remarks to the Author: Reproducibility):

Nothing to declare

Reviewer #3 (Remarks to the Author: Overall significance):

This manuscript describes a new implementation of Cas9 to generate *Aedes* mosquitoes that, it is hoped, can be released into wild populations as embryos, leading to elimination of those populations. Given the moniker “precision-guided sterility,” the rapid pace of theoretical and experimental improvements in Cas9 gene drive systems, and the senior author’s expertise in this realm, I expected this would describe a multi-component, controlled gene drive system. In actuality, it is a straightforward but well-designed use of Cas9 to generate mutations in two target genes, of which is required for male fertility and the other for female wing posture and therefore flight. Transgene lines that, when crossed to a Cas9-expressing line, gave nearly complete male sterility and female flightlessness were selected and analyzed. The use of the female flightlessness is an important addition that means mosquitoes will not need to be sex-sorted after production.

Reviewer #3 (Remarks to the Author: Impact):

The approach outlined here has some distinct advantages over release of sterile males in the RIDL approach and release of *Wolbachia*-infected males. It has the disadvantage of involving transgenic mosquitoes. It is unclear whether it will ultimately be successful in getting approval, but it is important that the method be published so it can be part of the discussion. I expect that publication of this work will lead to substantial discussion.

Reviewer #3 (Remarks to the Author: Strength of the claims):

The work is generally well documented and controlled, with appropriate rigor. There are a few issues that require additional discussion:

1. An important question that is not discussed fully concerns persistence of pgSIT in the environment. Fig 1B shows that there are rare males that are fertile. Scaling up to the numbers proposed for release

suggests that the probability that the transgene will be introduced into the local population is not zero. Of course, if the population is completely eliminated after several rounds of releases, then rare fertile males or flight-capable females are not a problem. However, this possibility (which is not a problem for RIDL, as it does not involve transgenic animals) likely does raise the bar on what probability of population elimination will be considered acceptable to obtain approvals.

2. I am not an expert in modeling, but MGD_{Drive} appears to have been developed for gene drive. Was this used previously to simulate successive releases, such as of Wolbachia-infected males?

In addition, there are a number of minor issues in the writing that need attention:

1. I was initially put off by the use of male and female symbols in text, but I will admit that it does make it easier to read. However, the use of σ 's (I used the male symbol here, in case it does not reproduce in final comments) is a possessive, not a plural. The standard genetic nomenclature for plural is to use two copies of the symbol adjacent to one another, as in $\sigma\sigma$ (two consecutive male symbols; this may need to be defined at first use). I really don't think the symbols work as superscripts.
2. line 80: "...we demonstrate resulting progeny of flightless ♀ 's and fit sterile σ 's that can compete...: is confusing, as it sounds like it is the progeny of flightless x sterile mosquitoes that can compete, rather than the progeny of the cross of the two strains, and that it is the sterile males that can compete for mating.
3. line 103: It is confusing to report 0-94% fertility and then say two lines reached 100% sterility. (Actually, sterility and fertility are not qualitative traits – an individual is one or the other. The quantitative trait is fecundity.)
4. line 105: 3 of 5 became a fraction
5. line 116: I don't think this is true introgression, just backcrossing. Also, "trans-homozygous" doesn't make sense as a word; these are double homozygotes (also in Fig. 2A).
6. line 120: This doesn't require a response unless it is an error, but it seems backwards to determine copy number and transgene location after building the double homozygote rather than before.
7. Fig. S1B: Proteins were compared but the legend uses italics and even says "betaTub gene."
8. Fig. S7: Panels need to be labeled.
9. Fig. S8 and S9: Panel B doesn't seem to be helpful. Is this just showing low coverage of ONP reads across this exon? There is no description of Oxford Nanopore sequencing of mutations provided (there is

only ONP of transgene insertion sites and copy number). “Target sites” is really gRNA targeting sequence. A line showing the actual cut site (at least on the strand shown) would help.

Reviewer #3 (Remarks to the Author: Reproducibility):

Statistical tests are appropriate and sufficient detail is given in Methods to allow reproduction (with the exception of the Oxford Nanopore sequencing mutations being missing).

Reviewer #4 (Remarks to the Author: Overall significance):

The Akbari laboratory has previously reported on an elegant CRISPR/Cas9 approach for population suppression of the fruit fly *Drosophila melanogaster*. This manuscript reports on the adaptation of this approach to the *Ae. aegypti* mosquito.

The work is technically sound, and adequately detailed. Comments follow.

- In Figure 2C, mating was allowed to occur for different lengths of time. However, missing is the time of the day at which each experiment was initiated, as mating is not uniform in any 24-hour period, as it is controlled by the circadian clock. The time of the day at which each experiment was initiated needs to be specified.
- Figure S4 is difficult to understand. There are boxes with arrows but some of the fluorescence is not visible. Also, the transmitted light images are often poor and it is not clear which organ is the source of fluorescence.
- Are the data in Fig. S5F the same as in Fig. 2B? If not, what is the difference?
- Important: whereas the modelling of Fig. 4 seems to be OK (but I am not an expert), it needs to be put into context in the Discussion. The island is very small, about 2 sq. miles. It is not clear if this figure is for the Teti'aroa atoll, or whether the Onetahi islet is even smaller. Please state the actual figure in the Discussion. The population is also very small (about 100 persons, is this correct??), and this should also be specified in the Discussion. To put the findings into context, the Discussion should comment on (i) the logistics to implement this strategy in a city where major dengue or Zika epidemics occur and (ii) given that this is a population reduction approach (as opposed to population elimination), the need of implementation of this strategy on a continuous basis (and possible costs) should be considered.

In summary, this manuscript adapts from *Drosophila*, an elegant strategy for suppression of *Aedes*

populations. The experiments are expertly executed, and the modelling is probably sound. However, considerations of what would it take to implement this strategy in a major endemic area should be clearly considered in the Discussion.

Reviewer #4 (Remarks to the Author: Impact):

This work offers an alternative strategy for mosquito population reduction, that has several advantages mentioned by the authors.

However, as pointed out, I feel it crucial that the possible shortcomings also need to be discussed, to provide a balanced view that the readers can evaluate.

Reviewer #4 (Remarks to the Author: Strength of the claims):

The work is technically sound.

Reviewer #4 (Remarks to the Author: Reproducibility):

I have no comments on the topic of reproducibility.

Author rebuttal, first version:

Reviewer 1

Reviewer #1 (Remarks to the Author: Overall significance):

High significance. More details in my attached summary.

Reviewer #1 (Remarks to the Author: Impact):

High impact. More details in my attached summary.

Reviewer #1 (Remarks to the Author: Strength of the claims):

Convincing and well-supported by data. The modeling work needs more information, as I mention in my one major critique in the attached summary.

Reviewer #1 (Remarks to the Author: Reproducibility):

I did not see any problem with the reproducibility or statistics.

Summary:

The authors present data describing the performance of their pgSIT approach to population suppression in the applied organism, *Ae. aegypti*. pgSIT provides a mechanism to generate sterile males without the use of irradiation. Two parental lines (one homozygous for Cas9 and one homozygous for sgRNAs targeting female flight and male fertility) are crossed to generate sterile males that can be released to compete with wild-type males to mate with wild-type females. In this manuscript, the authors **describe a complete story, from design and testing of genetic components to validate gene targets, construction of pgSIT mosquitoes to test their performance and fitness, extensive cage-based population suppression experiments, and simulation modeling for a real-world release scenario. The data is of very high quality and supports the conclusions drawn.** There are extensive and detailed results presented in the supplementary figures that allow interested readers to ‘dig in’, but the overall story presented is clear and easy to follow. The figures are well-organized with a few minor suggestions noted in my critiques below. The only major revision requested is to provide more details and justification for the parameters used during simulation modeling. **This paper is of very high impact.** While a proof of concept was previously published in flies, the demonstration of the technology in an applied insect species is of substantial impact, and is by no means trivial. My understanding of the Guided OA request to review is that the editors have already made a decision to pass on Nature Genetics and are only considering this for Nature Communications or Communications Biology. **I would request that the editors reconsider and instead publish this in Nature Biotechnology or Nature Genetics, which are a better fit because of the quality, impact, and widespread interest in this work.** There are suggested improvements listed beneath my signature. I recommend this paper for publication in Nature Communications with minor revisions. I welcome questions or concerns about the content of this review by the authors of the manuscript or editors. If it is against editorial policy to have non-anonymous reviews, please remove my name below.

Best,

Mike Smanski

smanski@umn.edu

Substantive points to consider during revision:

1. There are several details regarding the modeling that should be revised, better justified, or better explained.

First, please justify the use of 16 mosquitoes per human structure (and no mosquitoes outside of human structures) for the simulation. The total number of adult mosquitoes at equilibrium in this model is less than 1000. A previous modeling paper using

the same site (Strugarek et Dumont 2018) estimate the population of adult males on the islet to be greater than 5000 (so the total adult population would be ~10,000, or 10x higher than used in this study).

The paper mentioned (Strugarek et Dumont 2018) refers to population density of a different species of mosquito - *Aedes polynesisis* - not *Aedes aegypti* - which is the species we study in this manuscript. There really is not a good estimate of the current *Aedes aegypti* population density on the island of Onetahi, Tetiaroa, French Polynesia. A study from 2013, revealed that *Aedes aegypti* were significantly smaller (>20X less in some areas) than *Aedes polynesisis* populations on several islands in French Polynesia (1). Given these facts, and the very small size of Tetiaroa (~193 acres), which is roughly 10X smaller than the UCSD campus (~1976 acres), we believe an estimate of 16 mosquitoes per human structure is adequate. Moreover, the population size of *Ae. aegypti* is also consistent with density found in other locations (Sánchez C et al. 2020). Going forward, as we advance pgSIT to the field, we intend to directly quantify the population density and feed those parameters into models that can accurately predict the outcome of the intervention.

Second, it is unclear if the 400 pgSIT eggs released per adult mosquito are only male or are male plus female eggs. Please be more explicit. If it is male plus female, then presumably the pgSIT females will survive to adulthood and the subpanels in Figure 4B need to be updated to clarify that the y-axis is no. of wild-type adult female *Ae. aegypti*.

We apologize for the lack of clarity and have updated the text, previously “ ... 400 pgSIT eggs per wild adult ... “, to be more explicit, now stating “ ... 400 pgSIT eggs per wild adult (♂ and ♀) ...” Additionally, we have updated the figure legend to clarify that it plots wild-type adult females only and the methods to indicate that all calculations were performed for male and female adults.

Lastly, more details on egg hatch rate in the model should be included in the main text. The authors state that releasing 400:1 pgSIT:wild-type is similar to a 10:1 release ratio of adults, but this similarity depends a lot on the modeled survival rate of their released eggs (which presumably would be hatched from containers with greater survivability than wild laid eggs). If the 400 released eggs are all male, it would be most appropriate to describe this as a 800:1 release of pgSIT males to adult males.

Again, we apologize for the (too) concise description. We have updated the methods to include the following:

“Egg release size was calculated following the precedent in Kandul et al (<https://doi.org/10.1038/s41467-018-07964-7>): density-independent mortality rates for juvenile life stages were calculated for consistency with the population growth rate in the absence of density-dependent mortality, while density-dependent mortality was applied to the larval stage following Deredec et al (<https://doi.org/10.1073/pnas.1110717108>), and both rates were used to calculate the expected number of eggs required to match a 10:1 adult (♂ and ♀) release ratio.”

Minor points:

1. Line 77: Change Aegypti to aegypti.

Corrected

2. Line 104: Would be interesting to mention that the maternal Cas9 was better in all of the measurable comparisons (i.e those where both versions worked 100%).

We appreciate this suggestion. While this is generally true, there are a few cases where paternal Cas9 also resulted in complete sterility or 100% female flightless phenotypes (Table S3, Fig. S2 and S3) - therefore we cannot make this claim.

3. Line 598: Elsewhere (e.g. figures) ratios are given as pgSIT:Wild-type, while here the inverse is given. Better to keep it consistent and list these as 1:1, 5:1, 10:1, etc.

Great suggestion. Corrected

4. Figure 1: B and E: I get why you are showing both the maternal and paternal cas9/sgRNA on the right side of these panels, but it is still somewhat confusing why you are showing both types of crosses. You don't show the maternal and paternal for the other controls, and numbers for the maternal vs paternal for the two cas9+sgRNA are not distinct. It seems more straight forward to just show the compiled results and say "there was no difference when the Cas9 was paternal and the sgRNA maternal and vice versa". B and E: I think it would be better if you have a label to the left of the chromosome schematics, and label which chromosomes are shown (it seems like you drop Chr. 2, but this is not clear).

We have plotted all this data (Fig. 1B,E and Fig. 2E) for both maternal and paternal Cas9 in Fig S2, S3 and provided the raw data in Table S3 and Table S4. In the figure legend we also cite where all the data can be found, for example in the Fig 1. Legend we state "Data from both paternal Cas9 crosses (Cas9♂ x gRNA♀) and maternal Cas9 crosses (Cas9♀ x gRNA♂) are shown (Fig. S2, S3, Table S3)," and in the Fig. 2 legend we state " Bar graph comparing the percentage of transheterozygous and heterozygous Cas9 or gRNA progeny to those of WT (data can be found in Table S4, Video S4)." This will make it easy for the reader to dig in and find the data they are looking for. In terms of removing the maternal/paternal Cas9 data from Fig. 1B,E - we prefer to keep that data shown in the figure since we have the space. While we can just state that there was no difference - we prefer to show this since it is important - and all the control data is present in the supplement.

I think it would be better if you have a label to the left of the chromosome schematics, and label which chromosomes are shown (it seems like you drop Chr. 2, but this is not clear).

Chromosome labels are provided in both Fig 1 and Fig 2.

Legend: You refer to B and E as histograms, but these are better described as bar graphs with each datapoint shown. (conventional histograms show a probability distribution across the x-axis variable broken into bins).

Corrected.

5. Figure 2: B: the y-axis of progeny survival (%) does not really work since the grey bars are not showing survival. Better to just label the y-axis 'Percent', and keep the inset labels as they are. C: I think you mislabeled the pgSIT mosquitoes being taken out. It should say 250 instead of 50. Legend: same comment about histograms from above.

Corrected. Thank you for pointing this.

6. Figure 3: The eye colors in gRNA and pgSIT mosquitoes in the key in A are hard to see.

The mosquito colored eyes were increased in panel A for facilitate pgSIT / gRNA mosquito discrimination from wt and Cas9 mosquitoes..

7. Figure 4: The font and style for the subpanel labels A, B, C are different from the rest of the Figures.

We have adjusted the font type to match the other figures.

Reviewer 2

Reviewer #2 (Remarks to the Author: Overall significance):

The paper brings a very interesting approach on how to obtain completely sterile males and removal of 100% females. The title is misleading for something not reflected in the manuscript results. Half of the results are missing in the abstract section, with no mention about the flightless mosquitoes nor the combination of the two strategies.

Thank you for the suggestions. We have added the flightless females to the abstract. We prefer to keep the title as is since we did demonstrate population elimination using pgSIT.

For the figures, all legends are incomplete. And the figures are extremely hard to comprehend and pollute.

We have gone through and expanded many of the figure legends and modified some figures to help make them more understandable.

For the supplementary material should complement the manuscript and bring useful information, but the way is presented is that every single possible way to present something that sound as a result was included. And their majority could have been included in the text. This reflects a confusing thinking about the supplementary material selection. The main figures are so crowded that part of them should be the supplementary material, replacing most of the current ones. The manuscripts attempt to bring a fullness and grandiosity intimidating the readers and leading them to a mental confusion and accepting everything that was presented.

We appreciate the comment. The main results have been presented in 4 main figures and the auxiliary results are presented in the supplement. We have modified these figures according to reviewers' suggestions to make them easier to comprehend.

Lines 117-120: What is the potential reason for the other non-selected strains to have variable fertility levels and even full fertility? Same for flying females of non-selected pgSIT strain.

This is likely a result of position-effect-variegation (PEV) leading to reduced expression levels of the transgene at different genomic insertion sites. This is a very common phenomenon in insect transgenesis, and to mitigate this we generated many lines and selected for the best performing sites (read more here: https://en.wikipedia.org/wiki/Position-effect_variegation).

Reviewer #2 (Remarks to the Author: Strength of the claims):

Lines 125-128: Why authors were so confident that they would have superior results and “robust” transgene expression? Specially, when other strains obtained with the same construction are fertile (Figure S2) and some females even though they were flightless, some were still able to blood feed (Figure 3).

We developed many strains with different insertion sites (e.g. 10 lines for B2Tub and 5 lines for my-fem) and selected the insertions that performed the best. This experimental validation method enabled us to select the best performing strains to cross together. See point above regarding PEV.

Lines 132-134: Linked to the previous comment and the speculative sentence on these lines, the potential application of egg release for population suppression, and given the unrealistic possibility that these females may find conditions to mate, blood feed and lay eggs, how would be the potential impact of these offspring? Since regulatory bodies may find this information quite alarming.

We appreciate the comment. If flightless females mate to wildtype males, blood feed and lay eggs, the transgene may be introduced into the local population. However the chance is very-very low. First, *Aedes aegypti* males detect females via the sounds produced by female beating wings, and following a fast-paced, mid-air pursuit, attempt to seize and mate with them (2–5). Flightless females that were produced in our study can't fly (Fig 2D) and hold wings properly (Fig 1F), so they can't produce the mating sounds. Second, these flightless females move slowly and in a lethargic manner (supplement video 2), with significantly reduced survival (Fig S5A, B, Table S5), the chance to find blood source, take blood successfully, and then find oviposition sites is very low. Third, the releasing process will continue for several rounds, even if the transgene is introduced into the local population, the population will be completely eliminated after several rounds of releases. Finally, if needed a egg-release device (see Fig 1 in this rebuttal) could be utilized that would require the mosquitoes to fly out of the rearing container to reach the environment. This would further prevent the flightless females from entering the environment.

We envision pgSIT to be regulated in a similar manner to Oxitec's RIDL technology, which has been successfully deployed in many locations and recently received experimental use authorizations in the USA (<https://www.oxitec.com/florida>). Remember that the Oxitec RIDL technology was only 95% penetrant - meaning that females do get released that can bite and produce offspring - despite this - the system was approved for releases - it works and populations have been suppressed. We agree that the new technology may take several years to move from lab to nature application and it is unclear whether it will ultimately be successful in getting approval, but it is important that the method be published so it can be part of the discussion.

Lines 137-142: It's known that is not only time but also the moment that the coitus is interrupted so a second male can contribute with sperm to the offspring (DOI: 10.1371/journal.pone.0193164). By number used at the beginning of the experiment (which is also not clear the methods) the male sexual harassment over females may be very high, and within the 2-6 hours, males keep removing each other during coitus or several males trying to mate at same time with a flying female, causing the tendency of females not to move/fly to avoid coitus. The other possibility is the mating vigor (which is not the mating capacity as shown) and males need longer periods to mate.

We have provided the females the opportunity to mate with pgSIT males for 5 separate time periods using a conservative 1:5 (WT female : pgSIT) release regimen. This resulted in reduced fertility - which is a strong indication that our released males had mating vigor. The next step would be to take this system and compete with WT male in a continuous population -and that is what we aim to do in our next study. We have checked the methods section to make clear that the experimental design was clearly articulated.

Lines 151-154: If as stated in the manuscript, no significant impact on the fitness cost was found, however comparing these result with a study using ionizing gamma radiation on male mosquitoes, in both studies the population suppression took 6 generations to collapse using the same sterile : fertile ratio (DOI: 10.1098/rstb.2019.0808), and using eggs for releases indicates a higher impact on fitness by the use of much higher ratios (20 : 1 or 40 : 1) for the same number of generations to reach population suppression.

Thanks for pointing out another study which demonstrates the release of sterile males can rapidly suppress *Ae. aegypti* populations. This study uses a trivial technique to generate sterile males (gamma irradiation) which is not the same method we used in our manuscript - so not directly comparable. Moreover, even though our egg-release experiments were successful at suppressing the populations - we provide an explanation in the discussion as to why our discrete generation experimental design was not ideal for testing the egg release strategy :

“It should be noted that the releases of adult *pgSIT*[♂]'s unexpectedly resulted in faster population suppression as compared to egg releases in multigenerational population cage experiments. We believe this to result from the slightly reduced egg hatching rates of *pgSIT*[♂]'s and their delayed larva-pupa development time, which likely enabled the co-released WT ♂'s first access to WT ♀'s. While this could impact the discrete generation population cage experiments conducted here, it should not be problematic for suppressing continuous populations in the wild.”

Lines 164-167: As there were no field releases, why 25% reduction in male competitiveness was chosen? Another indication that the pgSIT males have lower fitness, is the comparison with a real field release in Brazil performing a mark-release-recapture experiment using drones and gamma irradiated males, in which within couple of releases (in the same period) they manage to observe more than 50% reduction of viable eggs in the field compared with the control site (DOI: 10.1126/scirobotics.aba6251).

Line 181: I am skeptical to call the strain 100% fit.

We appreciate the concern about fitness, but part of our reasoning for choosing a 25% reduction in mating fitness was to alleviate concerns about an “unobserved” fitness cost in the lab becoming relevant once the construct reaches field conditions. Additionally, it made an excellent midpoint for our sensitivity analysis (Fig. 4C). These are preliminary results ensuring that the minimum effectiveness of the current construct is capable of producing the desired results, which ensures that a (currently-not-seen) minor fitness reduction does not reduce the viability of this approach.

This design circumvents irradiation of mosquitoes, a deliberate choice to avoid fitness costs associated with irradiation.

Lines 181-183: The cage suppression experiment is based on a discrete and multigenerational process, which is quite artificial and not realistic. A similar experiment established a wild population in the semi-field cages, before starting releasing their “sterile” males. And because they also followed control cages during the study period, the replicates showed it is clear the impact and the low variability among the same sample group (DOI: 10.1038/s41586-019-1407-9).

We appreciate the comment. We have made very clear in both the results section and in the methods section that we have tested our mosquitoes using a discrete experimental design. We agree that semi-field experiments would be ideal, and this is going to be our next step that is well beyond the scope of this manuscript.

Lines 183-186: Excessive negative fitness cost can be a problem during operational releases, even in a small trial. The transgenic strain used in Brazil (OX513A) was known to have negative fitness cost (DOI: 10.1371/journal.pone.0020699 and DOI: 10.1371/journal.pone.0026086). And the outcome of this was seen overlapping the studies using this same strain for mass production and population suppression (DOI: 10.3791/3579 and DOI: 10.1371/journal.pntd.0003864). Although the authors claim to have achieved population suppression around 70 to more than 90%, the released area was quite small (5.5 ha with around 500 residences) and required a production of 0.5 million males/week to reach this result.

Thanks for the insight on these published papers. We do understand that fitness will be important to monitor during mass scale up and release - and ideally strains are selected that do not have major fitness issues. In our small-scale experiments we did achieve population suppression - the hope is that this could be repeated at scale in the future.

Line 188: It is written “realized” but do you mean “released”?

“Realized” is correct here.

Line 192: females produce 450 eggs throughout their life, this is a very high assumption since this would be around 90 eggs/gonotrophic cycle, which may not be realistic depending on strain, blood quality, oviposition site and other factors.

This is not an “assumption” - they can produce up to 450 eggs in their lifetime -a reference has been provided along with the # eggs per gonotrophic cycle. The key words “up to” are also included to imply that not all females will produce 450 eggs as there are many factors that will contribute to this output as the reviewer correctly points out.

Lines 198-199: Although field models show promising results using pgSIT, the scalability was not modeled and tested under realistic mass rearing facility to attend areas of pilot trials that are slightly more challenged than Onetahi island.

We appreciate the comment. We agree that the next step is to demonstrate scalability of the pgSIT system using a mass rearing facility - however these experiments are well beyond the scope of this manuscript.

Line 200: Senecio's technology so far is not available for operational trials, and Debug haven't tried transferring their technology to other operational trials with bigger populations and the maximum mentioned was less than 3 million males/week, which depending on the trial site is not enough to suppress the population.

We appreciate the comment and have verified that we have not made any unsubstantiated claims regarding the availability of Verily/Senecio's technologies.

Lines 204-207: Egg releases sound not as a promising technique since there will be no control on the male quality produced. And the dissemination of these eggs might be as laborious as adult production and release, unless the target is only to produce a product and not vector control. How would be the dynamics of deploying dissemination stations and how would be the entomological surveillance of these areas?

Egg-releases are being done by Oxitec with great success and reduced effort - so we don't envision pgSIT egg releases to be "as laborious." We have not worked out the dynamics of egg releases - but imagine these to be similar in design to Oxitec's approach (see figure 1 and read more: <https://www.oxitec.com/en/news/oxitec-successfully-completes-first-field-deployment-of-2nd-generation-friendly-aedes-aegypti-technology>). Moreover, for any mosquito intervention surveillance will need to be done before, during and post releases - and there are several traps that could be used for these purposes (e.g. BG Sentinel traps, Ovitraps).

Fig 1. Containers used for Oxitec V 2.0 fsRIDL egg releases in Florida (2021). Eggs are hatched in water and males fly out of the holes. A similar container can be used for pgSIT - requiring the males to FLY out - preventing any surviving flightless females (most die on the surface of the water due to lack of flight) from even entering the environment.

Here is a section of the discussion to address these points:

“This strategy will be especially effective for mosquitoes that diapause during the egg stage (e.g. *Aedes* species) because it will enable long-term egg accumulation. Eggs could be distributed to logistically spaced remote field sites where they can hatch, develop into adults, and fly out to compete with wild mosquitoes (Fig. S11). These hatching containers could be engineered in such a way to require the adults to fly out thereby preventing the release of flightless females. This attractive feature should reduce the costs of developing multiple production facilities requiring on-site sex separation for manual release of fragile adults.”

Line 201-214: However, the proposed model also showed that egg release was less effective as adult releases, meaning that the suppression delay will persist. Figure 4: According to some quick calculations using data from the manuscript only for Onetahi, which seems to be a quite isolated area and without a real mosquito burden and disease transmission, using the pgSIT it would be necessary to obtain 0.8 million eggs of the strain to be released every week. And a facility to keep this production that depends on

two strains to obtain, will definitely be a complex process since they must be independent facilities, multiplying by two the area, team, equipment all consumables in order to produce for a quite small trial.

As we acknowledged above (lines 151-154), our experimental design disadvantages egg releases compared to equivalent adult releases. However, simulations indicate that our presumption in the discussion is correct - there is still the slight delay that you point out but it does not impact overall suppression of the population. The delay is unfortunate, but the benefits of egg rearing and release reflect positively on the ability to scale this technology (see below).

The isolated nature of Onetahi acts as a natural containment feature for additional protection against unforeseen issues during trials. This is in accordance with suggested paths from design to use for novel gene drives (6).

We want to stress that we are not trivializing the difficulty of scaling a novel technology. The points you bring up are valid, and we do not have solutions for them yet. The benefits of this system are 1) it can make use of existing facilities without requiring novel safety measures, until the final generation when both pieces are combined, and 2) eggs can be stored for long periods of time, compared to adults, so the required mass can be generated over time. However, before a technology can be scaled for production, it must exist. We are at the existence stage, attempting to prove validity of this approach, and the next steps include larger efficacy trials and discussions on efficient scalability.

Reviewer 3

Reviewer #3 (Remarks to the Author: Overall significance):

This manuscript describes a new implementation of Cas9 to generate Aedes mosquitoes that, it is hoped, can be released into wild populations as embryos, leading to elimination of those populations. Given the moniker “precision-guided sterility,” the rapid pace of theoretical and experimental improvements in Cas9 gene drive systems, and the senior author’s expertise in this realm, I expected this would describe a multi-component, controlled gene drive system. In actuality, it is a straightforward but well-designed use of Cas9 to generate mutations in two target genes, of which is required for male fertility and the other for female wing posture and therefore flight. Transgene lines that, when crossed to a Cas9-expressing line, gave nearly complete male sterility and female flightlessness were selected and analyzed. The use of the female flightlessness is an important addition that means mosquitoes will not need to be sex-sorted after production.

Reviewer #3 (Remarks to the Author: Impact):

The approach outlined here has some distinct advantages over release of sterile males in the RIDL approach and release of Wolbachia-infected males. It has the disadvantage of involving transgenic mosquitoes. It is unclear whether it will ultimately be successful in getting approval, but it is important that the method be published so it can be part of the discussion. I expect that publication of this work will lead to substantial discussion.

Thank you. We agree 100%. :)

Reviewer #3 (Remarks to the Author: Strength of the claims):

The work is generally well documented and controlled, with appropriate rigor. There are a few issues that require additional discussion:

1. An important question that is not discussed fully concerns persistence of pgSIT in the environment. Fig 1B shows that there are rare males that are fertile. Scaling up to the numbers proposed for release suggests that the probability that the transgene will introduced into the local population is not zero. Of course, if the population is completely eliminated after several rounds of releases, then rare fertile males or flight-capable females are not a problem. However, this possibility (which is not a problem for RIDL, as it does not involve transgenic animals) likely does raise the bar on what probability of population elimination will be considered acceptable to obtain approvals.

In this study we were successful at generating two strains that when crossed produced 100% sterile males and flightless females (we counted >100,000 progeny so this is very robust). That said, it's possible that when this is scaled further to releasing millions or even billions there may be some error rate. What error rate is acceptable? What error rate will still enable suppression? We don't have these answers, however we can compare this system to Oxitec's RIDL system which was only >95 % penetrant:

“Penetrance of expression of the lethality trait is > 95% (i.e., 95% of the GE mosquitoes contain the lethality trait).” - quote from <https://www.fda.gov/files/animal%20&%20veterinary/published/Oxitec-Mosquito---Draft-Environmental-Assessment.pdf>.

This means that ~5% of the progeny that inherit the RIDL system survive - both females that can bite and transmit diseases and males/females that can mate out and further pass on the traits. Was this acceptable - yes these have been released into the environment all over the world. Did this error rate result in compromised success? Nope, RIDL has been demonstrated to be effective at suppressing >90% of the populations in the wild at release sites. Going forward we hope to further scale/test this system in the field to address these questions.

2. I am not an expert in modeling, but MGDriVE appears to have been developed for gene drive. Was this used previously to simulate successive releases, such as of *Wolbachia*-infected males?

While MGDriVE was designed for gene drives, thanks to the modular design, it is quite flexible at handling any inheritance pattern describable using Punnett squares (We have yet to find one that isn't). The most direct answer to your concern is this paper, <https://doi.org/10.1186/s12915-020-0759-9>, where *Wolbachia* was used for model and parameter validation, while the crux of the paper were reciprocal translocations and an underdominant system, UD^{MEL}.

In addition, there are number of minor issues in the writing that need attention:

1. I was initially put off by the use of male and female symbols in text, but I will admit that it does make it easier to read. However, the use of ♂'s (I used the male symbol here, in case it does not reproduce in final comments) is a possessive, not a plural. The standard genetic nomenclature for plural is to use two copies of the symbol adjacent to one another, as in ♂♂ (two consecutive male symbols; this may need to be defined at first use). I really don't think the symbols work as superscripts.

We appreciate the comment and are happy to make this modification if necessary. However, we would like some clarity from the editor of the journal beforehand to see what is preferred by the journal.

2. line 80: "...we demonstrate resulting progeny of flightless ♀'s and fit sterile ♂'s that can compete...: is confusing, as it sound like it is the progeny of flightless x sterile mosquitoes that can compete, rather than the progeny of the cross of the two strains, and that it is the sterile males can compete for mating.

Agreed and corrected.

3. line 103: It is confusing to report 0-94% fertility and then say two lines reached 100% sterility. (Actually, sterility and fertility are not qualitative traits – an individual is one or the other. The quantitative trait is fecundity.)

Agreed and corrected.

4. line 105: 3 of 5 became a fraction

Corrected

5. line 116: I don't think this is true introgression, just backcrossing. Also, "trans-homozygous" doesn't make sense as a word; these are double homozygotes (also in Fig. 2A).

Corrected

6. line 120: This doesn't require a response unless it is an error, but it seems backwards to determine copy number and transgene location after building the double homozygote rather than before.

This is correct. We backcrossed to WT to generate trans-heterozygotes then performed the analysis of copy number and transgene location.

7. Fig. S1B: Proteins were compared but the legend uses italics and even says "betaTub gene."

Corrected

8. Fig. S7: Panels need to be labeled.

Corrected

9. Fig. S8 and S9: Panel B doesn't seem to be helpful. Is this just showing low coverage of ONP reads across this exon? There is no description of Oxford Nanopore sequencing of mutations provided (there is only ONP of transgene insertion sites and copy number). "Target sites" is really gRNA targeting sequence. A line showing the actual cut site (at least on the strand shown) would help.

In Fig. S8 and S9, panel B refers to the ONP reads that align to this gene. If you look carefully you can see both low coverage and mutations in the DNA sequence. We agree that the panel is small - but find it useful and informative. In both these figures, the gRNA target sequence is shown in yellow and the direction of the gRNA target site is shown in the bottom of panel E. We agree that including the exact CRISPR cut sites would be great - but unfortunately these are not known. NHEJ produces random mutations near the gRNA target site - and these can include many kinds of Indels and span 10-20bp from the PAM (sometimes more). Therefore, the only sequence we can really highlight is the gRNA target sequence - and as expected mutations are found at both the DNA level (ONP) and the RNA level (RNAseq).

Reviewer #3 (Remarks to the Author: Reproducibility):

Statistical tests are appropriate and sufficient detail is given in Methods to allow reproduction (with the exception of the Oxford Nanopore sequencing mutations being missing).

We appreciate the comment. We have added a whole section in the methods dedicated to the Oxford Nanopore sequencing methods/analysis:

"To determine the transgene insertion site(s) and copy number(s), we performed Oxford Nanopore DNA sequencing. We extracted genomic DNA using the Blood & Cell Culture DNA Midi Kit (Qiagen, Cat# 13343) from twenty adult transheterozygous *pgSIT³*'s (3 days old) harboring all three transgenes (*Cas9*+/+ ; *gRNA^{βTub#7}* /+ ; *gRNA^{myo-fem#1}* /+), following the manufacturer's protocol. The sequencing library was prepared using the Oxford Nanopore SQK-LSK109 genomic library kit and sequenced on a single MinION flowcell (R9.4.1) for 72 hrs to generate an N50 read length for the set of 4088 bp. Basecalling was performed using ONT Guppy basecalling software version 4.4.1, generating 2.94 million reads above quality threshold $Q \geq 7$, which corresponds to 8.68 Gb of sequence data. To determine transgene copy number(s), reads were mapped to the AaegL5.0 reference genome (7) supplemented with transgene sequences (OA-1067A1: *gRNA^{βTub}*; OA-1067K: *gRNA^{myo-fem}*, and OA-874PA: Nup50-Cas9) using minimap2 (8). In total, 2,862,171 out of 2,936,275 reads (97.48%) were successfully mapped with a global genome-wide depth of coverage of 5.495. We calculated the mean coverage depth for all contigs in the genome (2310) and the three plasmids (OA-1067A1: *gRNA^{βTub}*; OA-1067K: *gRNA^{myo-fem}*; and OA-874PA: Nup50-Cas9) as well as normalized coverage (Table S9-S10).

Transgene coverage ranged from 5.1 to 7.6, and normalized coverage ranged from 0.93 to 1.38. As compared to the three chromosomes, the coverages are consistent with the transgenes present at a single copy (Fig. S7).

To identify transgene insertion sites, we inspected reads that aligned to the transgenes in the Interactive Genomics Viewer (IGV) browser. The reads extending beyond the boundaries of the transgenes were then analyzed to determine mapping sites within the genome. For OA-874PA, one read spanned the whole transgene (~11.5 kb) and extended 4 and 3.5 kb on both sides. The extending portions mapped to both sides of the position on NC_035109.1:33,210,105 (chromosome 3), with the nearest gene being AAEL023567, which is ~5 kb away. For OA-1067K, one read covered ~7 kb of the transgene extending ~10 kb off the 3' end, 9 kb of which map to the NC_035108.1:287,686-296,810 region (chromosome 2). A few other shorter reads map to the same location. The site is located in the intron of AAEL005206, which is a capon-like protein, and based on the RNA-seq data, its expression does not appear to be affected in pgSIT animals. For OA-1067A1, the nanopore sequencing was unable to resolve the insertion site, presumably due to its insertion in one of the remaining gaps in the genome. Finally, using nanopore data, we confirmed genomic deletions in both pgSIT target genes - see AEL019894 and AAEL005656 as expected (Fig. S8-S9). The nanopore sequencing data has been deposited to the NCBI sequence read archive (SRA) under BioProject ID is PRJNA699282 with accession number SRR13622000."

Reviewer 4

Reviewer #4 (Remarks to the Author: Overall significance):

The Akbari laboratory has previously reported on an elegant CRISPR/Cas9 approach for population suppression of the fruit fly *Drosophila melanogaster*. This manuscript reports on the adaptation of this approach to the *Ae. aegypti* mosquito. The work is technically sound, and adequately detailed. Comments follow.

- In Figure 2C, mating was allowed to occur for different lengths of time. However, missing is the time of the day at which each experiment was initiated, as mating is not uniform in any 24-hour period, as it is controlled by the circadian clock. The time of the day at which each experiment was initiated needs to be specified.

We appreciate this comment. We agree with the review that mating is not uniform in any 24-hour period, as it is controlled by the circadian clock. We have added the time of the day in the Material and Method.

“To determine whether prior matings with *pgSIT*[♂] could reduce ♀ fertility, we initiated 15 cages each consisting of 250 mature (4–5 days old) *pgSIT*[♂] combined with 50 mature (4–5 days old) WT virgin ♀. We allowed the *pgSIT*[♂]s to mate with these ♀s for a limited period of time (including: 2, 6, 12, 24, and 48 hrs, all experiment start from 9:00 am PST, 3 replicate cages each)

- Figure S4 is difficult to understand. There are boxes with arrows but some of the fluorescence is not visible. Also, the transmitted light images are often poor and it is not clear which organ is the source of fluorescence.

We appreciate this comment. This supplemental figure was a significant undertaking (240 merged images!!!). Overall, we believe the images depict the phenotypes quite nicely and to make it easier to understand we have now incorporated a key at the bottom to define what the arrows represent

- Are the data in Fig. S5F the same as in Fig. 2B? If not, what is the difference?

No, Fig. 2B compares the flight ability and fertility of trans-heterozygous mosquitoes to those in WT and heterozygous mosquitoes. Groups of 10 or 50 mosquitoes were used to generate these data (fewer data points, Table S4). Fig. S5F shows the fitness data on the fertility of homozygous parental lines (Cas9 and gRNA) as compared to WT and trans-heterozygous mosquitoes. Individual mosquitoes were used for the assessment of fitness data (many data points), Table S5.

- Important: whereas the modelling of Fig. 4 seems to be OK (but I am not an expert), it needs to be put into context in the Discussion. The island is very small, about 2 sq. miles. It is not clear if this figure is for the Teti'aroa atoll, or whether the Onetahi islet is even smaller. Please state the actual figure in the Discussion.

Thanks for this comment, these releases were simulated on the motu of Onetahi, Teti'aroa, French Polynesia. This is stated several times in the text. Yes, the atoll Teti'aroa is ~2.2 mi². The motu Onetahi is approx. 73.8 hectare (https://www.stat.auckland.ac.nz/~jrusell/files/rep/Tetiarao_Jul09_en.pdf). We added this figure to the Fig. 4 legend.

The population is also very small (about 100 persons, is this correct??), and this should also be specified in the Discussion. To put the findings into context, the Discussion should comment on (i) the logistics to implement this strategy in a city where major dengue or Zika epidemics occur and (ii) given that this is a population reduction approach (as opposed to population elimination), the need of implementation of this strategy on a continuous basis (and possible costs) should be considered.

Thank you for the comment, we have expanded our discussion to incorporate these suggestions:

“For pgSIT to be realized in the wild, the two strains will first need to be separately and continuously mass-reared in a facility, without contamination, and crossed to produce sterile ♂’s. While this can be viewed as rate-limiting²¹, it offers stability, as the binary CRISPR system will remain inactive until crossed—thereby reducing the evolution of suppressors or mutations that could disrupt the system. Additionally, each sorted ♀ can produce up to 450 eggs in her lifetime (~90 eggs per gonotrophic cycle)²², which improves scalability. Moreover, once crossed, the resulting progeny are essentially dead-ends (i.e. sterile ♂’s /flightless ♀’s), hatched among high numbers of sterile *pgSIT*³’s, and should not contribute to the gene pool²³. We demonstrate here that the technology is fully penetrant by screening >100K individuals.

pgSIT offers an alternative approach to scalability that should help decrease costs and increase efficiency. For instance, the required genetic cross at scale can be initiated using existing robotic sex sorting devices (www.senecio-robotics.com) or⁵. Upon sex sorting and crossing, the resulting *pgSIT* progeny can be distributed and released at any life stage, mitigating requirements for sex separation at field sites. This strategy will be especially effective for mosquitoes that diapause during the egg stage (e.g. *Aedes* species) because it will enable long-term egg accumulation. Eggs could be distributed to logistically spaced remote field sites where they can hatch, develop into adults, and fly out to compete with wild mosquitoes (Fig. S11). These hatching containers could be engineered in such

a way to require the adults to fly out thereby preventing the release of flightless females. This attractive feature should reduce the costs of developing multiple production facilities requiring on-site sex separation for manual release of fragile adults. That said, to achieve adequate control in major urban settings, repeated releases will likely be required on a continuous basis and costs will need to be considered.”

In summary, this manuscript adapts from *Drosophila*, an elegant strategy for suppression of *Aedes* populations. The experiments are expertly executed, and the modelling is probably sound. However, considerations of what would it take to implement this strategy in a major endemic area should be clearly considered in the Discussion.

Thanks for your comment, we have provided this in the discussion:

“Finally, notwithstanding its inherently safe nature, pgSIT requires genetic modification, and regulatory use authorizations will need to be granted prior to implementation. While this could be viewed as a limitation (9), we don’t expect obtaining such authorizations to be insurmountable. In fact, we envision pgSIT to be regulated in a similar manner to Oxitec’s RIDL technology, which has been successfully deployed in many locations and recently received experimental use authorizations in the USA.”

Reviewer #4 (Remarks to the Author: Impact):

This work offers an alternative strategy for mosquito population reduction, that has several advantages mentioned by the authors.

However, as pointed out, I feel it crucial that the possible shortcomings also need to be discussed, to provide a balanced view that the readers can evaluate.

Thank you for this comment, we have discussed a few shortcomings in the discussion to provide a balanced view. These include:

“For pgSIT to be realized in the wild, the two strains will first need to be separately and continuously mass-reared in a facility, without contamination, and crossed to produce sterile ♂’s. While this can be viewed as rate-limiting (9), it offers stability, as the binary CRISPR system will remain inactive until crossed—thereby reducing the evolution of suppressors or mutations that could disrupt the system.”

And

“Finally, notwithstanding its inherently safe nature, pgSIT requires genetic modification, and regulatory use authorizations will need to be granted prior to implementation. While this could be viewed as a limitation (9), we don’t expect obtaining such authorizations to be insurmountable.”

Reviewer #4 (Remarks to the Author: Strength of the claims):

The work is technically sound.

Reviewer #4 (Remarks to the Author: Reproducibility):

I have no comments on the topic of reproducibility.

References

1. L. K. Hapairai, *et al.*, Field evaluation of selected traps and lures for monitoring the filarial and arbovirus vector, *Aedes polynesiensis* (Diptera: Culicidae), in French Polynesia. *J. Med. Entomol.* **50**, 731–739 (2013).
2. L. M. Roth, A Study of Mosquito Behavior. An Experimental Laboratory Study of the Sexual Behavior of *Aedes aegypti* (Linnaeus). *American Midland Naturalist* **40**, 265 (1948).
3. G. Wishart, G. R. van Sickle, D. F. Riordan, Orientation of the Males of *Aedes aegypti* (L.) (Diptera: Culicidae) to Sound. *The Canadian Entomologist* **94**, 613–626 (1962).
4. P. Belton, Attraction of male mosquitoes to sound. *J. Am. Mosq. Control Assoc.* **10**, 297–301 (1994).
5. A. Aldersley, L. J. Cator, Female resistance and harmonic convergence influence male mating success in *Aedes aegypti*. *Sci. Rep.* **9**, 2145 (2019).
6. K. C. Long, *et al.*, Core commitments for field trials of gene drive organisms. *Science* **370**, 1417–1419 (2020).
7. B. J. Matthews, *et al.*, Improved reference genome of *Aedes aegypti* informs arbovirus vector control. *Nature* **563**, 501–507 (2018).

8. H. Li, Minimap2: pairwise alignment for nucleotide sequences. *Bioinformatics* **34**, 3094–3100 (2018).
9. J. Bouyer, M. J. B. Vreysen, Concerns about the feasibility of using “precision guided sterile males” to control insects. *Nature Communications* **10** (2019).

Reviewer comments, second version:

Reviewer #1 (Remarks to the Author: Overall significance):

[See original review, as the significance did not change in revision]

Reviewer #1 (Remarks to the Author: Impact):

[See original review, as the impact did not change during revision]

Reviewer #1 (Remarks to the Author: Strength of the claims):

This work is convincing, as the main aim of this study is to demonstrate the first proof of concept of pgSIT in mosquitoes. The authors rigorously demonstrate that the pgSIT approach is working well in mosquitoes and the efficacy (penetrance) is remarkably high.

Some minor points of correction to the revised document: The authors attempted to clarify that both male and female eggs were released in the simulation, but the revised version is still unclear. I recommend changing to "up to 400 pgSIT eggs (male and female) per wild adult were simulated...". Also, the authors changed the number from 400 to 200, both in the Results and in the Figure legend. The y-axis in the figure goes from 0 to 400, so I am not sure why that change was made during revision. I believe it should be switched back to 400. Lastly, the *Ae. aegypti* on the y-axis of Figure 4 should be italicized.

Reviewer #1 (Remarks to the Author: Reproducibility):

[See original review, as the reproducibility did not change during revision]

Reviewer #3 (Remarks to the Author: Overall significance):

This manuscript describes a new implementation of Cas9 to generate sterile *Aedes* mosquitoes for population eliminate. It is a well-designed use of Cas9 in the laboratory/production site to generate, in one cross, mutations in two target genes. One of these is required for male fertility and the other for female wing posture and flight. This should be preferable to the current irradiation scheme. The use of the female flightlessness is an important addition that means mosquitoes will not need to be sex-sorted after production.

Reviewer #3 (Remarks to the Author: Strength of the claims):

The relatively minor deficits in the first submission have been addressed, resulting a stronger and more complete manuscript. No additional problems were noted.

Reviewer #3 (Remarks to the Author: Reproducibility):

Statistical tests are appropriate and sufficient detail is given in Methods to allow reproduction.

Reviewer #4 (Remarks to the Author: Overall significance):

The authors responded satisfactorily to most of my comments. However, the authors artfully circumvented the issue of commenting on what it would take to implement this strategy in a major dengue-affected city.

In the manuscript, the model gives the number of eggs needed to be released weekly the motu of Onetahi, to achieve population suppression.

In my original comment “the logistics to implement this strategy in a city where major dengue or Zika epidemics occur”, I meant that the scale of the project needs to be considered. Please take a major city of your choice that is prone to dengue epidemics, and provide an estimate of the weekly number of eggs that need to be deployed to effectively suppress the *Ae. aegypti* population in that city. If an approximate cost to produce these eggs can be estimated, this would be very helpful.

Author rebuttal, second version:

Reviewer #1 (Remarks to the Author: Overall significance):

[See original review, as the significance did not change in revision]

Reviewer #1 (Remarks to the Author: Impact):

[See original review, as the impact did not change during revision]

Reviewer #1 (Remarks to the Author: Strength of the claims):

This work is convincing, as the main aim of this study is to demonstrate the first proof of concept of pgSIT in mosquitoes. The authors rigorously demonstrate that the pgSIT approach is working well in mosquitoes and the efficacy (penetrance) is remarkably high.

Some minor points of correction to the revised document: The authors attempted to clarify that both male and female eggs were released in the simulation, but the revised version is still unclear. I recommend changing to "up to 400 pgSIT eggs (male and female) per wild adult were simulated...". Also, the authors changed the number from 400 to 200, both in the Results and in the Figure legend. The y-axis in the figure goes from 0 to 400, so I am not sure why that change was made during revision. I believe it should be switched back to 400. Lastly, the *Ae. aegypti* on the y-axis of Figure 4 should be italicized.

We thank the reviewer for catching this error. The reviewer is correct that the number of pgSIT eggs (male and female) released per wild adult (male and female) is up to 400. We have corrected both the Results and Figure legend accordingly. Additionally, the *Ae. aegypti* on the y-axis of Figure 4 has been italicized, as per the reviewer's good advice.

Reviewer #1 (Remarks to the Author: Reproducibility):

[See original review, as the reproducibility did not change during revision]

Reviewer #3

Reviewer #3 (Remarks to the Author: Overall significance):

This manuscript describes a new implementation of Cas9 to generate sterile *Aedes* mosquitoes for population elimination. It is a well-designed use of Cas9 in the laboratory/production site to generate, in one cross, mutations in two target genes. One of these is required for male fertility and the other for female wing posture and flight. This should be preferable to the current irradiation scheme. The use of the female flightlessness is an important addition that means mosquitoes will not need to be sex-sorted after production.

Reviewer #3 (Remarks to the Author: Strength of the claims):

The relatively minor deficits in the first submission have been addressed, resulting in a stronger and more complete manuscript. No additional problems were noted.

Reviewer #3 (Remarks to the Author: Reproducibility):

Statistical tests are appropriate and sufficient detail is given in Methods to allow reproduction.

We thank this reviewer for all the constructive feedback that has strengthened our manuscript.

Reviewer # 4

Reviewer #4 (Remarks to the Author: Overall significance):

The authors responded satisfactorily to most of my comments. However, the authors artfully circumvented the issue of commenting on what it would take to implement this strategy in a major dengue-affected city. In the manuscript, the model gives the number of eggs needed to be released weekly the motu of Onetahi, to achieve population suppression. In my original comment “the logistics to implement this strategy in a city where major dengue or Zika epidemics occur”, I meant that the scale of the project needs to be considered. Please take a major city of your choice that is prone to dengue epidemics, and provide an estimate of the weekly number of eggs that need to be deployed to effectively suppress the *Ae. aegypti* population in that city. If an approximate cost to produce these eggs can be estimated, this would be very helpful.

Editor - We agree with Referee #4 that a concrete example of scalability should be included in the manuscript, with a detailed overview of eggs per week (and ideally cost), for further consideration by Nature Communications.

We appreciate the reviewer’s and editor’s request for a concrete example of scalability to be included in the manuscript, including an overview of eggs per week. We consider Pape’ete, the capital of French Polynesia, to be a suitable case study for a location, as it is a major dengue-affected city (<https://www.iamat.org/country/french-polynesia/risk/dengue>), is comparable to Onetahi due to its location, and could conceivably be considered as an intervention site following a successful trial at a smaller trial location in French Polynesia.

Regarding egg release requirements, the simulations in Onetahi considered 62 human structures with 16 adult *Aedes aegypti* mosquitoes each, resulting in a total of 992 adult mosquitoes at the site. Simulated population elimination was possible for large yet achievable release schemes, such as 24 weekly releases of 100 or more *pgSIT* eggs. For the case of Onetahi, this would mean 24 weekly releases of 99,200 *pgSIT* eggs. The population of Pape’ete in the most recent census (2017) was 26,926 (<https://www.insee.fr/fr/statistiques/3294364?sommaire=2122700&q=populations+l%C3%A9gales+polyn%C3%A9sie+2017>) and the average household size in French Polynesia is 3.9 (<https://www.prb.org/international/indicator/hh-size-av/table/>). This suggests an approximate number of households in Pape’ete of 6,904. Assuming, as for Onetahi, 16 adult *Ae. aegypti* per household, this suggests a total wild adult *Ae. aegypti* population for Pape’ete of approximately 110,464. And in line with Onetahi again, this suggests elimination of *Ae. aegypti* would be conceivable for 24 weekly releases of 11,046,400 *pgSIT* eggs. While this is a large number of released eggs, we note that the release ratio (100 eggs per 1 wild adult) is comparable to (or perhaps smaller than) release ratios for other SIT programs, which commonly have adult release ratios of 10:1 and female *Ae. aegypti* produce >30 eggs per day in temperate climates (as mentioned in the manuscript). We have added the following text describing the approximate expected release requirements in Pape’ete in the Results and Discussion sections of the manuscript:

Results: “For Onetahi, weekly releases of 100 or more *pgSIT* eggs per wild *Ae. aegypti* adult translate to weekly releases of ~100,000 *pgSIT* eggs. Scaling this up to a dengue-affected city such as Pape’ete, French Polynesia with ~100 times as many housing structures, this would require weekly releases of ~10,000,000 *pgSIT* eggs.”

Discussion: “This translates to weekly releases of ~100-200 thousand *pgSIT* eggs in Onetahi, and releases of ~10-20 million *pgSIT* eggs in a city such as Pape’ete.”

We do not feel we are in a position to estimate costs at this stage, however, as these do not scale linearly with release size and must include a range of factors including the production facility, community engagement, materials and labor. We plan to rigorously explore cost-effectiveness in subsequent analyses in the coming years.